# Evolution of a distinct chromatin regulatory landscape in brown algae

Jeromine Vigneau[1,3], Jaruwatana Sodai Lotharukpong [1,3], Pengfei Liu [1], Remy Luthringer[1], Bérangère Lombard [2], Damarys Loew [2], Fabian B. Haas[1], Michael Borg [1]✉ & Susana M. Coelho [1]✉

Chromatin structure plays a central role in regulating transcription, genome stability and epigenetic inheritance in eukaryotes. Much of our understanding of chromatin architecture and histone post-translational modifications (hPTMs) comes from a narrow set of animal and plant models, but emerging data from non-model lineages are challenging canonical views of how chromatin functions across the tree of life. Brown algae are complex multicellular eukaryotes that provide a unique perspective on chromatin evolution given their independent origin of complex multicellularity. Here we compile the chromatin toolkit of brown algae and show that canonical silencing systems involving DNA cytosine methylation and Polycomb repressive complex 2 (PRC2)-mediated histone H3 lysine 27 (H3K27) methylation were lost early in their evolution. By generating hPTM profiles from diverse brown algal clades, we resolve the nature and regulatory roles of chromatin states in this lineage and show how H3 lysine 79 (H3K79) methylation emerged and diversified as a repressive system. We further uncover sex-specific reconfigurations in species with varying degrees of sexual dimorphism and reconstruct the ancestral regulatory landscape that probably preceded the emergence of brown algae. Together, our findings illuminate the dynamic evolution of chromatin regulation in a distinct multicellular lineage and challenge assumptions about the universality of chromatin-based mechanisms across eukaryotes.

In eukaryotes, chromatin regulates access to the genome by packaging DNA into an organized structure that modulates transcription and other DNA-based processes. This organization is defined by chromatin states, which consist of specific combinations of DNA and/or histone post-translational modifications (hPTMs), DNA-binding proteins and three-dimensional structural features[1–3]. Chromatin states are often dynamic and responsive to developmental and environmental cues, and in some cases can be stably maintained though epigenetic inheritance[4]. In addition to gene regulation, chromatin plays a critical role in safeguarding genome integrity by repressing transposable elements and other invasive mobile elements[5].

Most of our understanding of epigenetic and chromatin-based regulation has largely been informed by the discovery and analysis of hPTMs in a limited range of plant and animal models[6]. While many hPTMs are evolutionarily conserved across eukaryotic lineages and trace back to the last eukaryotic common ancestor, recent work on non-model lineages is revealing a surprising diversity of chromatin systems and challenges classical views about the conserved function of hPTMs[7]. For example, the brown alga *Ectocarpus* lacks canonical hPTMs such as histone H3 lysine 9 (H3K9) and lysine 27 (H3K27) methylation, while H3 lysine 79 (H3K79) methylation appears to play a repressive role rather than associating with active transcription as in

[1]Department of Algal Development and Evolution, Max Planck Institute for Biology Tübingen, Tübingen, Germany. [2]Institut Curie, PSL Research University, CurieCoreTech Mass Spectrometry Proteomics, Paris, France. [3]These authors contributed equally: Jeromine Vigneau, Jaruwatana Sodai Lotharukpong. ✉e-mail: michael.borg@tuebingen.mpg.de; susana.coelho@tuebingen.mpg.de

yeast and metazoans, highlighting a striking divergence in chromatin regulation[8–11]. The brown algae represent a distinct branch of complex multicellular eukaryotes, separated from animals and plants, that have evolved within the past 450 Myr and display a broad diversity in genome size, morphology and sexual systems[12–14]. With the recent development of genomic tools for this clade, including over 65 genome assemblies of which several are at a chromosome scale, as well as genetic tools, brown algae have emerged as attractive model organisms for comparative studies. Notably, their genomes have remained largely syntenic[15], which facilitates cross-species comparative genomic studies.

Here we investigated the diversity of chromatin landscapes across major brown algal lineages that encompass the broad morphological complexity and variation in sexual systems within the group, along with an outgroup species for comparison. We combined genome-wide chromatin modification maps with gene expression data and various genomic features to functionally interpret chromatin states across brown algae. Our analyses reveal several key insights into the evolution of chromatin landscapes in this group. First, the emergence of the brown algal lineage was accompanied by the loss of both Polycomb repressive complex 2 (PRC2)-mediated repression and DNA methylation, representing a major shift in epigenetic regulation during early brown algal evolution. Second, activation-associated hPTMs are highly conserved across species, suggesting functional constraint. In contrast, the repressive role of H3K79 methylation, which we show is common across multiple brown algae, appears to be an ancestral feature given it also marks repressed genes in their closest non-phaeophycean outgroup. Third, sex-specific chromatin state differences between male and female gametophytes seem independent from the degree of sexual dimorphism observed at the organismal level, suggesting that morphological sex differentiation may be driven by chromatin reconfiguration at a limited number of high-order effector loci. Finally, by examining chromatin and DNA methylation in an outgroup species, we trace the transition from an ancestral epigenetic landscape to the distinct architecture observed in brown algae.

## Results

### Evolution of chromatin- and epigenetic-related genes

The recent availability of several high-quality brown algal genomes allowed us to investigate the conservation and evolution of chromatin-associated proteins across the clade[12,15]. Using BLAST searches and orthology-based approaches, we screened for homologues of known chromatin-related proteins and revealed a conserved and distinct repertoire across brown algae (Fig. 1a). Aside from transfer RNA methyltransferases homologous to DNMT2 (TRDMT), we observed a complete absence of DNA methyltransferases across the clade, including DNMT1, DNMT3 and DNMT5 (Fig. 1a and Extended Data Fig. 1). DNMT1 orthologues were only identified in the closest non-phaeophycean relative Schizocladia ischiensis (Fig. 1a and Extended Data Fig. 1), suggesting that the lineage-specific loss of DNA methyltransferases occurred early in brown algal evolution. Similarly, we found that EED and SUZ12 subunits specific to PRC2 are absent from all surveyed brown algal species, while the modular MSI1 subunit common to other chromatin complexes was present. EED and SUZ12 orthologues are also absent in S. ischiensis as well as other closely related Ochrophytina species, suggesting that the loss of PRC2 may have preceded the emergence of the Phaeophyceae. The loss of PRC2 was also reflected in the absence of PRC1 homologues, which form a distinct repressive complex in animals and plants[16]. SET domain proteins are abundant across the clade, which included homologues of Trithorax-related histone methyltransferases and two brown algal-specific families of SET domain proteins. Interestingly, orthologues of DOT1, which is responsible for H3K79 methylation in yeast and animals[17], are greatly expanded among brown algae, highlighting a Phaeophyceae-specific adaptation in chromatin regulation (Fig. 1a). In addition to an ancestral clade (DOT1.1) that retains homology with yeast

DOT1 and human DOT1L, these histone lysine methyltransferases have diversified in brown algae into at least four additional subfamilies, each having novel domain associations and variable domain architectures (Extended Data Fig. 1). Differences in expression of DOT1 genes across the Ectocarpus life cycle further supports functional diversification among the expanded subfamilies (Extended Data Fig. 1). These observations show that the emergence of the brown algal lineage is marked by the concurrent loss of epigenetic control via PRC2 and DNA methylation and an expansion of DOT1 histone methyltransferases, suggesting a major shift in gene regulatory strategies in this clade.

To investigate the diversity of chromatin landscapes in brown algae, we selected a set of representative species spanning the phylogenetic breadth of the group, chosen to reflect variation in morphological complexity, sexual systems and reproductive strategies (Fig. 1b,c). To enable comparisons between male and female gametophytes within the same genetic background, we used sibling samples where possible, thereby minimizing genetic variability unrelated to sex (Supplementary Table 1). Our sampling included Desmarestia dudresnayi, which recently transitioned to co-sexuality (monoicy), alongside its closest relative Desmarestia herbacea, a dioicous species with separate male and female individuals. We also included Undaria pinnatifida, a kelp species characterized by a highly complex morphology, high sexual dimorphism and a large genome with expanded UV sex chromosomes[15]. This contrasts with Ectocarpus sp. 7 and Scytosiphon promiscuus, which possess smaller sex-linked regions on their UV sex chromosomes and have low-to-medium sexual dimorphism, respectively[15,18,19]. Finally, we included the filamentous chromista S. ischiensis as a representative outgroup species from the closest diverging lineage outside the brown algae[20].

We first performed mass spectrometry of histone preparations to detect hPTMs across the five representative species, as done previously in Ectocarpus[9] (Fig. 1d, and Supplementary Tables 2 and 3). Overall, hPTMs were detected with minimal species-specific differences. Histones that we identified as H2A.Z-like variants carry heavily acetylated tails that are present in Ectocarpus, U. pinnatifida, D. herbacea and S. ischiensis, but were not retrieved in S. promiscuus and D. dudresnayi. H3K9me1 was detected in D. herbacea and S. ischiensis, and H3K9me3 in U. pinnatifida, but neither mark was observed in the other species (Fig. 1d and Supplementary Table 2). Importantly, methylated forms of histone H3K27 were absent from all five brown algal species analysed, consistent with the lack of PRC2 subunits encoded in their genomes (Fig. 1a,d and Supplementary Table 2). The lack of H3K9 and H3K27 methylation does not appear to be caused by mutations at or around these residues, given strong conservation of the N-terminal tails of histone H3 in brown algae compared with Arabidopsis H3 variants (Supplementary Fig. 1). Taken together, these data indicate that the lack of canonical epigenetic regulation via DNA and H3K27 methylation is a general feature of brown algae.

In earlier studies focusing on chromatin landscapes in the filamentous brown alga Ectocarpus, we identified H3K4me3, H3K9ac and H3K36me3 as hPTMs of active chromatin, whereas H3K79me2 and H4K20me3 were more strongly linked to transcriptional repression[8,9]. Except for H3K4me3 and H4K20me3, which were technically challenging to isolate in our mass spectrometry runs, we were able to detect most of the other hPTMs in each species (Supplementary Table 2). We further validated the presence of H3K4me3, H3K9ac and H3K36me3 in each species using immunoblotting, although H3K79me2 was only detectable in U. pinnatifida (Fig. 1d and Supplementary Fig. 1). We speculate that H3K79me2 is probably present at low levels and is challenging to detect with immunoblotting, which was further supported by similarly low signals in mouse histone extracts (Supplementary Fig. 1). Building on their conservation across the clade and prior characterization in Ectocarpus, we focused on this set of five hPTMs to investigate the evolutionary dynamics of chromatin landscapes in brown algae.

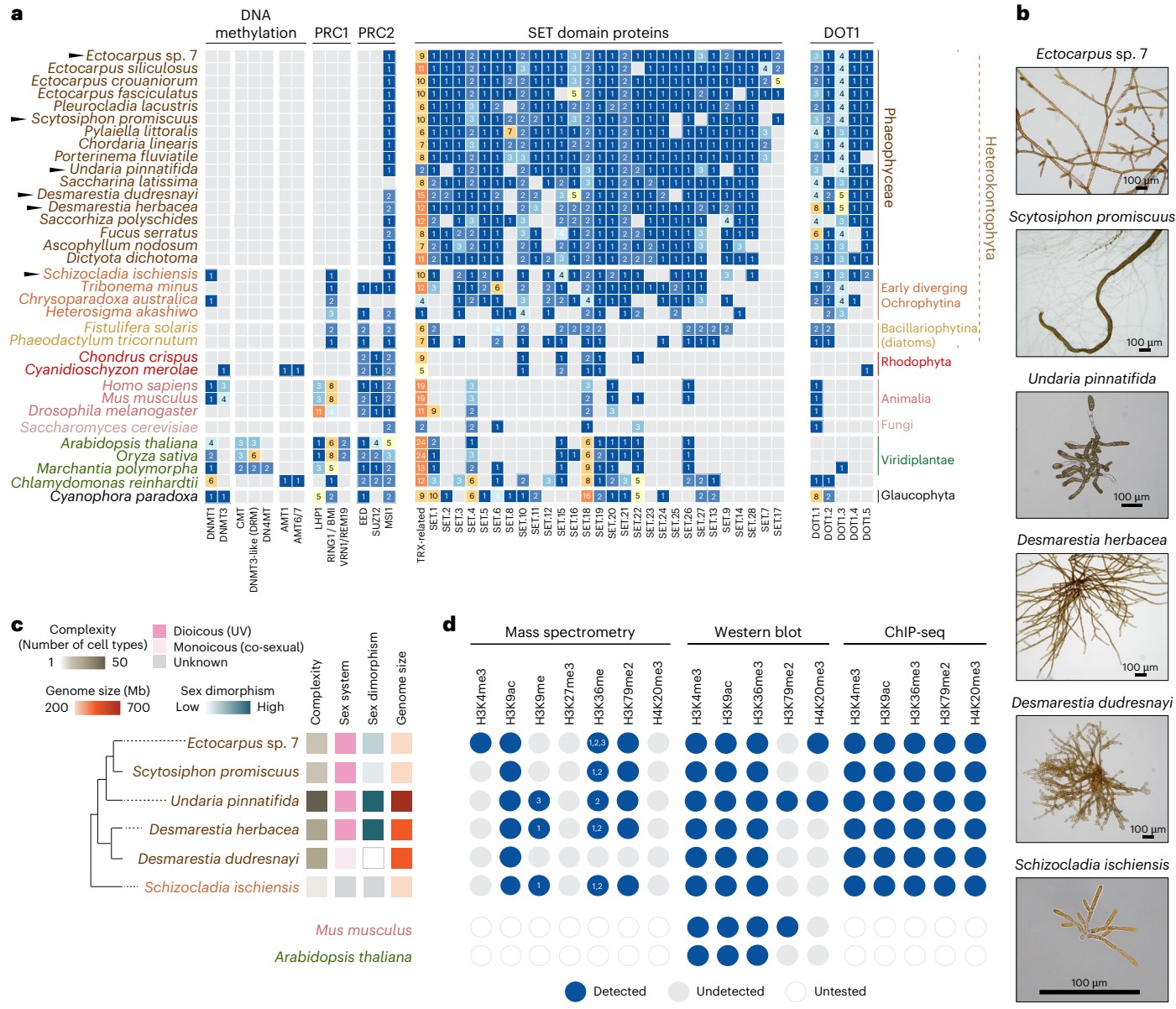

**Fig. 1 | Evolution of chromatin-related and epigenetic pathways in brown algae. a**, Census of chromatin proteins across the brown algae, their closest relatives and other major eukaryotic lineages. The total number of proteins belonging to a given gene family (or orthogroup) are shown, with low to high numbers colour coded from blue to orange. The species focused on in this study are indicated by a black triangle. **b**, Representative images of the species used in this study. **c**, Schematic diagram summarizing the phylogenetic relationship of the six species studied alongside biological features that distinguish them, namely their overall morphological complexity, sexual system, sexual dimorphism and relative genome size. **d**, Summary of the major hPTMs detected in the panel of six species using mass spectrometry, western blot and/or ChIP-seq. *Ectocarpus* mass spectrometry data were produced in a previous study[8].

## The chromatin landscape of diverse brown macroalgae

Closely related male and female gametophyte (haploid) lines were used to generate sex-specific chromatin immunoprecipitation sequencing (ChIP-seq) profiles for the five hPTMs in species with separate sexes (*Ectocarpus*, *S. promiscuus*, *U. pinnatifida*, *D. herbacea*) (Figs. 1b and 2a, Supplementary Table 1, and Methods). Monoicous *D. dudrenayi* gametophytes and vegetative tissue from *S. ischiensis* were similarly profiled (Figs. 1b and 2a). For each species, we profiled at least 400 clonal individuals per male, female, monoicous or vegetative replicate, then merged ChIP-seq replicates after confirming high reproducibility (Supplementary Fig. 2 and Supplementary Table 4).

Metaplot analyses revealed that H3K4me3 and H3K9ac were highly enriched at translational start sites (TSSs) and strongly associated with gene expression across all species (Fig. 2a and Extended Data Fig. 2). H3K36me3 was consistently enriched over the body of expressed genes.

H3K79me2 was also broadly deposited across gene bodies, marking approximately 25–50% of genes depending on the species (Fig. 2e and Extended Data Fig. 2). In *Ectocarpus* and *S. promiscuus*, H3K79me2 deposition along gene bodies was anticorrelated with higher transcript abundance, a pattern that was less pronounced in *U. pinnatifida* and the two *Desmarestia* species (Fig. 2e and Extended Data Fig. 2). Nevertheless, across all five brown algae, expressed genes consistently showed a clear depletion of H3K79me2 around the TSS (Extended Data Fig. 2). Indeed, genes with H3K79me2 at the TSS had significantly lower expression levels than unmarked genes (Fig. 2f and Extended Data Fig. 3). Moreover, genes with H3K79me2-marked TSSs were significantly enriched among genes with lower expression (quantiles 1–2) or genes with no expression at all (quantile 0), whereas genes in the highest expression quantiles were significantly enriched among those lacking the mark (Fig. 2g and Supplementary Table 5). The repression-associated mark H4K20me3

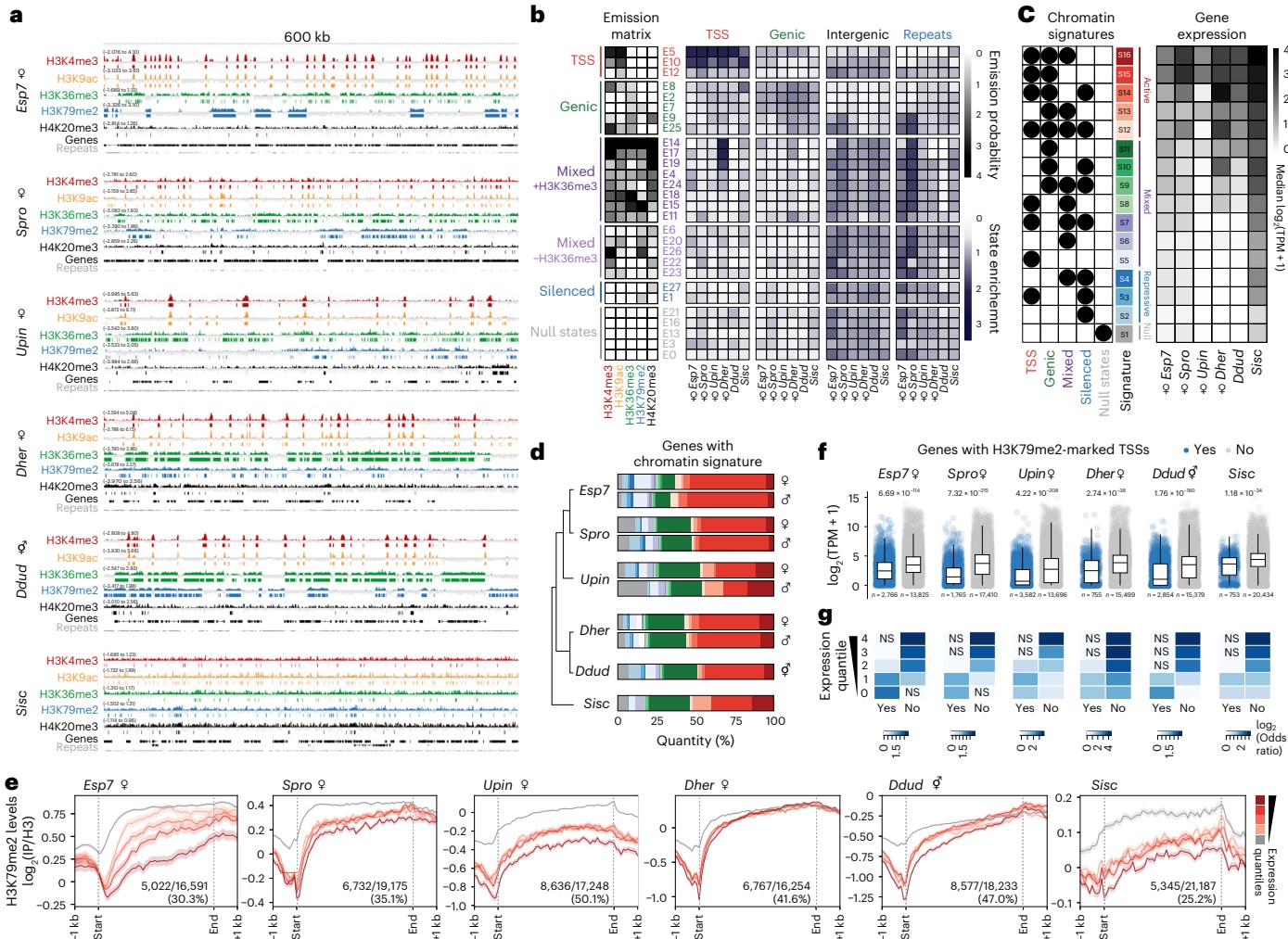

**Fig. 2 | The chromatin landscape across multiple brown algae.**
**a**, Representative genome browser tracks of ChIP-seq datasets in each species. Data for female gametophyte are shown for dioicous species. ChIP-seq coverage is represented as the $\log_2$ ratio of immunoprecipitation (IP) DNA relative to histone H3, with the range indicated on each track. ChIP-seq peaks for each hPTM are indicated under their respective track (see Extended Data Fig. 3a for male data). **b**, A model of chromatin emission states inferred by hiHMM across all ChIP-seq datasets (left) alongside the enrichment of each chromatin state in genomic features of each species (right). Female samples are shown for dioicous species (see Extended Data Fig. 3b for male data). **c**, Matrix of the chromatin signatures assigned to genes based on the hiHMM emission state model (Methods) summarizing the proportion of repressive, mixed and active chromatin states within each signature. The right panel displays the median RNA-seq expression level ($\log_2$(TPM + 1)) of genes associated with each signature. Female samples are shown for dioicous species (see Extended Data Fig. 3c for male data). **d**, Proportion of chromatin signatures assigned to genes in each sample. **e**, Metaplot of H3K79me2 levels over genes grouped into quantiles of gene expression (high to low; dark red to light red). Genes with no expression (TPM = 0) are also shown (grey). Gene bodies are plotted as proportional lengths, upstream and downstream intergenic regions in kilobases (kb). The metaplots are restricted to genes that overlapped with an H3K79me2 peak. Shading represents the standard error of the H3K79me2 enrichment signals. **f**, Expression of genes with (blue) or without (grey) an H3K79me2 peak overlapping their TSS in each species. Female samples are shown for dioicous species (see Extended Data Fig. 3d,e for male data). $P$ values were computed via the unpaired two-tailed Wilcoxon test. The boxplot shows medians (centre line), interquartile ranges (boxes) and whiskers extending to the most extreme values within 1.5× the interquartile range. **g**, Association between H3K79me2-marked TSSs and gene expression quantiles. Genes with TPM > 0 were assigned into quantiles of increasing expression (1–4), whereas genes with no detectable expression (TPM = 0) were assigned to category 0. Genes with and without H3K79me2-marked TSS are indicated as 'yes' and 'no', respectively. Significant overlaps between sets (adjusted $P$ values < 0.05; one-sided Fisher's exact test with Benjamini–Hochberg correction) are coloured by the $\log_2$(odds ratio). NS, non-significant overlaps.

also showed a clear depletion at the TSS of expressed genes across the five brown algal species (Extended Data Fig. 2). While H4K20me3 displayed a difference at the TSS between males and females in *Ectocarpus*, its relatively weak enrichment signal prevents definitive conclusions about this difference (Supplementary Table 4). In the outgroup species *S. ischiensis*, hPTMs showed similar deposition patterns although enrichment signals were weaker overall, with the strongest signals observed for H3K9ac at the TSS (Extended Data Fig. 2). Only a small subset of *S. ischiensis* genes were marked with H3K79me2 (753/21,187; 3.5%), but these nevertheless showed significantly lower expression than unmarked genes (Fig. 2f,g).

Next, we employed the Bayesian non-parametric framework hiHMM to jointly infer chromatin state maps across the six different species[21] (Supplementary Dataset 1; https://doi.org/10.17617/3.TDGYHS). This analysis produced a probabilistic model composed of 27 emission states (Fig. 2b and Supplementary Fig. 3), which we grouped into five broad categories based on their enrichment in specific genomic features: three TSS-associated states, five genic states, 14 mixed states distinguished by the presence or absence of H3K36me3, two silenced states, and five 'null' states lacking any of the assayed hPTMs (Fig. 2b and Extended Data Fig. 3). Closer inspection revealed both conserved and species-specific patterns of chromatin

state occurrence (Extended Data Fig. 4). For example, TSSs in all species were defined by a highly conserved and limited combination of states enriched for H3K4me3 and H3K9ac (E5, E10, E12). Gene bodies were predominantly characterized by three genic states common to all six species (E8, E2, E7). In contrast, intergenic regions and repeats exhibited more variable patterns that included a combination of mixed and silenced states. Notably, two genic states (E9, E25) were exclusively enriched in intergenic regions and repeats in *Ectocarpus* and *S. promiscuus*, suggesting a more derived chromatin landscape in the Ectocarpales lineage.

Multiple emission states can occur along the length of a single gene in various combinations, making it challenging to analyse chromatin state dynamics at the gene level. To address this, we applied our previously established approach to determine the presence of the five broad chromatin state categories at each gene across all species, resulting in 16 distinct combinations of chromatin signatures (Methods, Fig. 2c, Extended Data Fig. 3 and Supplementary Tables 6–11). Based on the predominant hPTMs associated with each chromatin state, we further classified the chromatin signatures into four main groups: active (S12–S16), mixed (S5–S11), repressive (S2–S4) and null (S1) (Fig. 2c and Extended Data Fig. 3). The relative distribution of chromatin signatures assigned to genes was broadly conserved among species, with active signatures S15 and S16 representing the most common categories (Fig. 2d). We verified the relationship between the 16 chromatin signatures and gene expression in each species using paired RNA sequencing (RNA-seq) data generated from the same biological material used for ChIP-seq profiling (Fig. 2c, Extended Data Figs. 3 and 5, Supplementary Fig. 4, and Supplementary Table 4). Across all brown algal species, genes assigned to active signatures consistently had higher transcript levels than those with mixed signatures, while genes associated with repressive or null signatures showed the lowest levels overall (Fig. 2c and Extended Data Fig. 5). In *U. pinnatifida* females, we noted that genes assigned to the active chromatin signatures S12 and S14 showed no detectable expression, which happened to also include H3K79me2 (Fig. 2c). However, the proportion of genes assigned to S12 and S14 is extremely small across all species (Fig. 2d), particularly in *U. pinnatifida* (S12: 343 in male (2.0%) and 355 in female (2.0%); S14: 29 in male (0.2%) and 209 in female (1.2%); 17,501 genes in total). This indicates that these chromatin signatures represent rare or ephemeral hPTM combinations rather than biologically stable categories that regulate transcription. In the *U. pinnatifida* male, H3K79me2 profiles were also notably different from those of the female, with a substantially elevated proportion of the null signature S1 overall compared with other datasets (Fig. 2d,e and Extended Data Fig. 3). We think that this pattern may be due to a mixed chromatin profile given that the *U. pinnatifida* male gametophyte sample we processed was highly fertile with an abundance of male gametes. In contrast, the outgroup *S. ischiensis* exhibited a weaker correlation between chromatin signatures and gene expression, particularly for repressive signatures, suggesting other regulatory systems may contribute to gene repression in addition to hPTMs.

Taken together, our findings indicate that hPTMs associated with active transcription function similarly as in other eukaryotic lineages, while H3K79me2 deposition appears to be consistently associated with transcriptional repression across brown algae. Moreover, given that H3K79me2 marks a subset of repressed genes in *S. ischiensis*, we posit that its repressive role probably predated the brown algal lineage and subsequently diversified both in its mechanism of action and in the breadth of genes it targets over the course of Phaeophyceae evolution.

## Conserved function of chromatin signatures among brown algae

The brown algae are thought to have emerged around 450 Myr ago, making them one of the youngest complex multicellular lineages, a fact that is reflected by the strong synteny observed across their genomes[12,22]. This prompted us to examine the evolutionary dynamics of chromatin signatures across the six species. For this, we first identified 3,143 conserved single-copy orthologues and assessed whether they retained similar signatures, using female data for species with separate sexes[8] (Supplementary Table 12). Our analysis revealed a strong overall conservation of chromatin signatures, which significantly exceeded expectation based on permutation tests assuming no conservation (Fig. 3a). Pairwise comparisons with model alga *Ectocarpus* further showed that the conservation of chromatin signatures correlated, in part, with phylogenetic distance (Fig. 3b). In *Ectocarpus*, these single-copy orthologues were strongly enriched for the transcriptionally active signature S15 and largely corresponded to deeply conserved genes with essential cellular functions and broad expression patterns, suggesting that they primarily represent housekeeping genes (Supplementary Fig. 5). These results highlight how orthologous genes have maintained similar active chromatin organization across brown algal evolution, which is probably driven by their constitutive expression and core housekeeping function.

We next asked whether chromatin signatures were associated with genes of similar evolutionary age and expression profile[23]. We used genomic phylostratigraphy to infer the evolutionary age of genes then assessed the distribution of their assigned signature, revealing statistically significant differences (Fig. 3c and Extended Data Fig. 6; Kruskal–Wallis test: $P < 2.2 \times 10^{-16}$ for all species). Notably, null and repressive chromatin signatures were more strongly associated with evolutionarily younger genes in *Ectocarpus* and across the other species (Fig. 3c and Supplementary Fig. 11). Moreover, when limiting our analysis to species-specific orphan genes, we found that these were strongly enriched in chromatin signatures with reduced expression (S1, S2, S5, S6, S8, S11) both in *Ectocarpus* and across the clade (Figs. 2c and 3d, and Extended Data Fig. 7). This suggests that younger genes typically reside in heterochromatic regions when compared with older and more conserved genes. Differences in chromatin signatures also manifested in similar expression dynamics. Genes with null and repressive chromatin signatures exhibited more restricted expression profiles than those with active signatures, a pattern that was consistent across all species, indicating that these signatures may represent facultative heterochromatin potentially involved in developmental gene regulation (Fig. 3e and Extended Data Fig. 8). Taken together, our findings demonstrate that chromatin signatures function similarly across 450 Myr of brown algal evolution[13], revealing conserved chromatin features underlying gene regulation in this lineage.

## The chromatin landscape involved in sex determination and differentiation

UV sex chromosomes have distinctive genomic and evolutionary features due to their mode of inheritance and their characteristic sex-determining regions (SDRs) that do not recombine[22,24,25]. In *Ectocarpus*, these features are reflected by a unique chromatin landscape on the UV chromosomes, where a much higher proportion of genes are marked by repressive chromatin signatures than on autosomes[8,10]. Strikingly, this pattern was highly conserved across all five brown algal species we examined, with sex chromosome genes consistently showing reduced proportions of active signatures and increased proportions of null and repressive signatures than autosomal genes (Fig. 4a). The UV sex chromosomes also consistently displayed a markedly lower proportion of conserved chromatin signatures than their autosomal counterparts (Fig. 4b). These findings show that the UV sex chromosomes have a distinct chromatin organization across brown algae, which reflects the rapid gene turnover that is characteristic of these fast-evolving chromosomes.

We next assessed chromatin dynamics involved in sexual differentiation by comparing chromatin signatures between males and females. To capture these dynamics, we focused on major chromatin reconfigurations by comparing shifts between the four chromatin

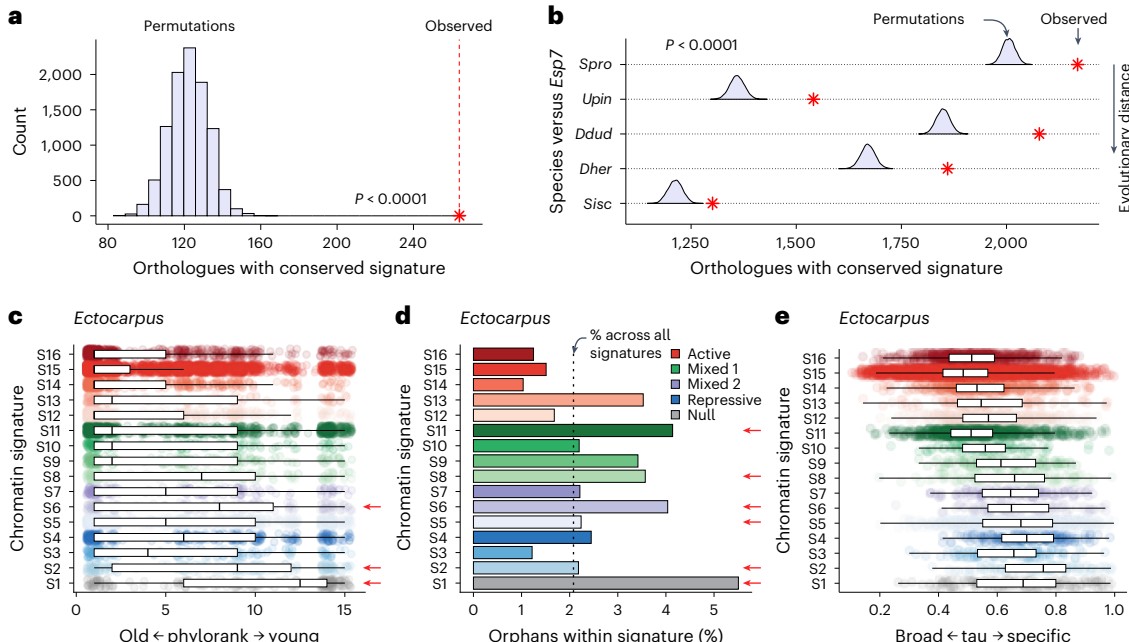

**Fig. 3 | Conservation and phylostratigraphy of chromatin signatures across brown algae. a**, Number of single-copy orthologues with the same chromatin signature across all six species, alongside permutations assuming no conservation. **b**, Number of orthologues with the same chromatin signature in pairwise comparisons of each individual species against the model alga *Ectocarpus*, alongside permutations assuming no conservation. **c**, Distribution of gene age (phylorank) across chromatin signatures in *Ectocarpus*. Using pairwise sequence alignment, genes with homologue(s) detected at the deepest taxonomic node (that is, cellular organisms) are assigned a phylorank of 1, whereas genes without any detectable homologues in any other species are given the phylorank of 15, which corresponds to the total number of taxonomic nodes defined for the species. **d**, Percentage of orphan genes assigned to each chromatin signature in *Ectocarpus*. Genes from phylorank 15 (species-specific) are assigned as orphan genes. **e**, Distribution of expression specificity scores (tau) for genes assigned to each chromatin signature in *Ectocarpus*. *P* values were calculated using a one-sided permutation test based on 10,000 permutations in **a** and **b**. Gene models with weak transcriptomic evidence (mean TPM < 1) were excluded from **c** and **d**, resulting in 14,926 genes with both expression evidence and reliable gene age estimates. Tau values were calculated for 11,433 genes from previously published data[60]. The boxplots show medians (centre line), interquartile ranges (boxes) and whiskers extending to the most extreme values within 1.5× the interquartile range.

signature groups (active, mixed, repressive and null). Across the four dioicous species, the vast majority of genes (63.2–87.8%) retained the same chromatin signature group between sexes (Fig. 4c). Of those that were dynamic between sexes, sex-biased genes (that is, genes that were differentially expressed between sexes) were significantly more likely to undergo major reconfiguration in *Ectocarpus* and *S. promiscuus* compared with the genomic background (Fig. 4d). In *D. herbacea* and *U. pinnatifida*, only female-biased genes were enriched for such changes (Fig. 4d). Genes located in the pseudoautosomal region (PAR) of the UV chromosomes were significantly enriched for dynamic chromatin reconfigurations in *Ectocarpus* and *S. promiscuus* (Fig. 4d). Taken together, these results suggest that major chromatin reconfigurations underlying sex-biased gene regulation are subtle and variable across brown algae and are largely restricted to specific loci, particularly female-biased genes.

Next, we focused on chromatin reconfigurations associated with the transition from separate sexes to co-sexuality by comparing male and female gametophytes of the dioicous species *D. herbacea* with monoicous gametophytes of *D. dudresnayi*. Similar to the dioicous species, the vast majority of genes retained the same chromatin signature group between the co-sexual and either sex of *D. herbacea* (Fig. 4e). As observed during sexual differentiation in the dioicous species, chromatin reconfigurations were preferentially associated with orthologous female-biased genes, whereas orthologous male-biased genes showed no dynamic differences in the co-sexuals compared with females, and were even less dynamic when compared with males (Fig. 4f). These results suggest that major chromatin changes underlying co-sexuality largely occur at female-biased genes, paralleling the situation observed in dioicy. Finally, we asked whether the transition

to co-sexuality influences the conservation of chromatin signatures on the UV sex chromosomes by testing whether the reduced conservation underlying dioicy is retained when a former sex chromosome becomes an autosome (hereafter termed the sex homologue[22]). Intriguingly, the *D. dudresnayi* sex homologue also showed reduced conservation of chromatin signatures, indicating that this property of sex chromosomes persists after the transition to monoicy (Fig. 4g). Thus, the former *D. dudresnayi* sex chromosome retains molecular footprints of its past life as a sex chromosome and can leave lasting evolutionary imprints at the chromatin level.

## Epigenetic control in the sister group of the brown algae

The outgroup *S. ischiensis* stood out among the six species we analysed due to its weaker correlation between chromatin signatures and gene expression, and its relatively low number of H3K79me2-marked genes. This prompted us to explore other potential regulatory mechanisms that may operate in *S. ischiensis*. Our phylogenetic analysis revealed that, unlike brown algae, *S. ischiensis* encodes a DNMT1 orthologue (Fig. 1a), leading us to speculate that DNA methylation may regulate gene expression in this species. To explore this, we called different forms of DNA methylation from Oxford Nanopore long-read sequences used to assemble the *S. ischiensis* genome. Consistent with the lack of 4-methylcytosine (4mC) and 6-methyladenine (6mA) methyltransferases in *S. ischiensis* (Fig. 1a), we observed very low (<5%) genome-wide levels of both 4mC and 6mA that were comparable to those detected on the plastid genome, suggesting that these are likely to be false positive signals (Extended Data Fig. 9). In contrast, N5-methylcytosine in a CG context (5mCG) was highly abundant, reaching almost 80% genome-wide and far exceeding background levels on the plastid

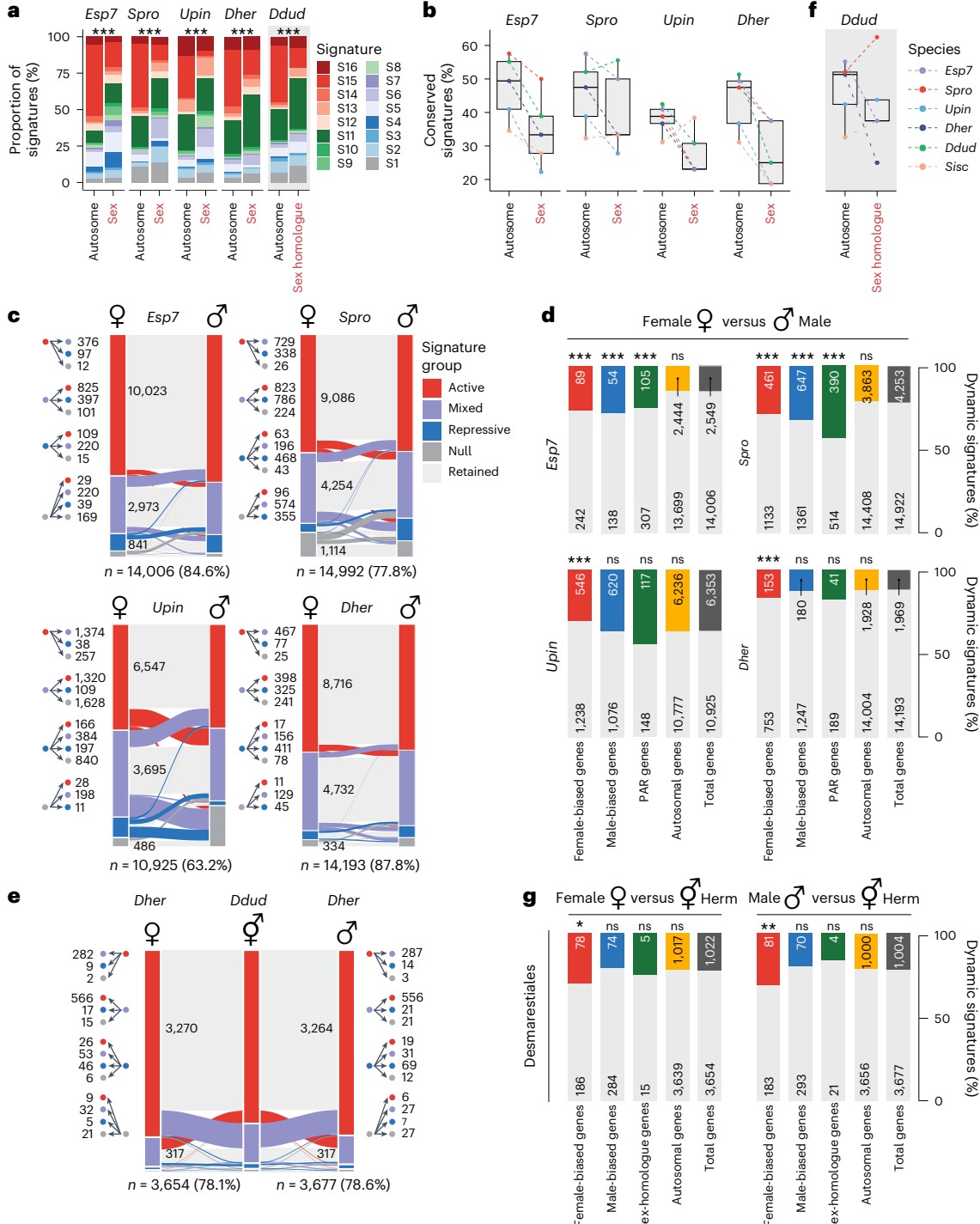

**Fig. 4 | Chromatin reconfiguration during sexual differentiation in brown algae with contrasting sexual systems. a**, Proportion of chromatin signatures on autosomes versus sex chromosomes across dioecious brown algal species. *P* values of a $\chi^2$ test, corrected for multiple comparison by the Benjamini–Hochberg procedure, on top of each bar indicate whether the proportions observed differ significantly between autosomes and sex chromosomes (or sex chromosome homologue in *Ddud*). The exact adjusted *P* values are: *Esp7*, $1.32 \times 10^{-56}$; *Spro*, $3.00 \times 10^{-203}$; *Upin*, $1.41 \times 10^{-55}$; *Dher*, $5.70 \times 10^{-12}$; *Ddud*, $2.92 \times 10^{-26}$. **b**, Chromatin signature conservation between autosomes and sex chromosomes in each species. **c**, Reconfiguration in chromatin signatures between females and males. The direction of chromatin signature changes and the number of genes involved are shown on the left, with the total number of genes shown below. Data table is available in Supplementary Table 14a. **d**, Proportion of sex-biased, PAR and autosomal genes that dynamically switch chromatin signature between sexes. *P* values of a $\chi^2$ test, corrected

by Bonferroni method, on top of each bar indicate whether the proportions observed differ significantly from the genome average. Data table is available in Supplementary Table 14b. **e**, Reconfiguration in chromatin signatures between females and males versus co-sexual *Desmarestia* species. The direction of chromatin signature changes and the number of genes involved are shown on the left, with the total number of genes shown below. Data table is available in Supplementary Table 14a. (**f**) Proportion of sex-biased, PAR and autosomal genes that dynamically switch chromatin signatures in *D. herbacea* (dioicous) versus *D. dudresnayi* (monoicous). *P* values of a $\chi^2$ test, corrected by Bonferroni method, on top of each bar indicate whether the proportions observed differ significantly from the genome average. Data table is available in Supplementary Table 14c. **g**, Chromatin signature conservation across all autosomes versus the sex homologue (ancestral sex chromosome) in the co-sexual species *Desmarestia dudresnayi*. *P* values were taken from the $\chi^2$ test. ns, not significant; *$P < 0.05$; **$P < 0.01$; ***$P < 0.001$.

genome (Fig. 5a,b). Although 5mC in CHG and CHH contexts also exceeded plastid levels, they showed no obvious enrichment pattern across the genome nor preference for transposable elements as they do in plants, suggesting that methylation in these contexts is unlikely to have a specific function in *S. ischiensis* (Fig. 5a and Extended Data Fig. 9).

Closer inspection of 5mCG in *S. ischiensis* revealed discrete regions with a complete loss of methylation around the promoter of protein-coding genes (Fig. 5a), which is reminiscent of unmethylated CpG 'islands' found in mammalian genomes[26,27]. These demethylated islands coincided with the enrichment of TSS-associated H3K4me3 and H3K9ac, with the upstream region of genes consistently displaying reduced 5mCG levels relative to surrounding genomic regions (Fig. 5a,b). Hierarchical clustering of 5mCG levels alongside the six profiled hPTMs revealed two main clusters distinguished by the presence or absence of promoter demethylation and the concurrent enrichment of active H3K4me3 and H3K9ac marks (Fig. 5c). Genes in this 'active' cluster showed higher transcript levels than those without promoter demethylation, consistent with the deposition of active hPTMs (Fig. 5d). In addition to these promoter-associated differences, we also observed a modest elevation of both 5mCG and H3K36me3 across the gene bodies of cluster 1 genes (Fig. 5c). However, 5mCG increase was not uniform across expression quantiles and did not scale with transcriptional output (Fig. 5e), indicating that while gene body methylation may be present, it is not strongly coupled to expression level. Cluster 2 genes, by contrast, showed no enrichment of 5mCG nor H3K36me3 within gene bodies relative to flanking regions. These results suggest that promoter demethylation may have played a key regulatory role prior to the loss of DNA methylation during early brown algal evolution.

## Discussion

Brown algae are the most recent lineage to have independently evolved complex multicellularity, making them an important case study to understand how regulatory mechanisms evolve during the emergence of organismal complexity. Our comparative analysis across the Phaeophyceae and their closest relatives reveals that this transition was accompanied by a fundamental shift in chromatin regulation, where canonical repressive pathways involving PRC2, H3K9 and DNA cytosine methylation are largely dispensable across the clade. Although the presence of DNA methylation is highly variable across eukaryotes[28], H3K27 methylation is largely ubiquitous and essential in animals, fungi, land plants and even closely related stramenopiles[29,30]. The concurrent loss of DNA, H3K9 and H3K27 methylation in brown algae is unprecedented among complex multicellular eukaryotes and underscores the unique evolutionary trajectory of this lineage.

Our analysis of the outgroup *S. ischiensis* provides an important perspective on the ancestral regulatory landscape of the brown algae. We show that the presence of a single DNMT1 enzyme is accompanied by high genome-wide 5mCG levels, highlighting yet another eukaryotic lineage that independently evolved hypermethylation[26,27,31,32]. Interestingly, we reveal a strong link between promoter demethylation and the accumulation of active histone marks at expressed genes, suggesting a regulatory model in which DNA methylation contributes to gene repression. The localized loss of 5mCG at promoters, together with the enrichment of active histone marks, points to a regulatory architecture in which promoter hypomethylation plays a central role. In line with observations from other eukaryotes, H3K4me3 may itself protect these promoter regions from DNA methylation[33,34], providing a plausible mechanism for the maintenance of these unmethylated islands. *S. ischiensis* also shows a modest increase in gene body 5mCG at expressed genes, although this does not scale with transcriptional output as in animals and plants[35,36]. This pattern was accompanied by a modest increase in H3K36me3, which is reminiscent of the coupling between H3K36me and gene body methylation in both vertebrates and invertebrates, where H3K36me directly recruits DNMT3 enzymes for de novo methylation[37–39]. However, DNMT3/DRM and

DNMT5 homologues are absent in *S. ischiensis*, suggesting that the apparent gene body methylation in *S. ischiensis* is mediated through a distinct mechanism, potentially involving DNMT1 itself. Thus, while these features differ from classical methylation systems in plants and animals, they nonetheless point to a lineage-specific configuration of DNA methylation and the possible convergent evolution of rudimentary gene body methylation. Finally, the absence of PRC2 orthologues in *S. ischiensis* suggests that H3K27me3-mediated regulation is likely to have been lost early in brown algal evolution. The loss of both DNA methylation and PRC2 in the common ancestor of brown algae, which is exceptionally rare among multicellular eukaryotes, would have necessitated the emergence of alternative repressive systems, providing fertile ground for the adaptation of DOT1-mediated H3K79 methylation into a key repressive pathway.

Our data suggest that the expansion and diversification of DOT1-like enzymes into lineage-specific families is a key feature of Phaeophyceae evolution. In yeast and animals, DOT1-dependent H3K79 methylation regulates diverse processes including gene expression, replication initiation, DNA damage response, microtubule reorganization and chromosome segregation[17]. DOT1 enzymes also promote heterochromatin formation by regulating pericentromeric transcription of satellite repeats, where bursts of transcription are required to establish and maintain long-term silencing[40,41]. We speculate that this role in heterochromatin formation could have been co-opted during brown algal evolution to give rise to its repressive role in the modern day. H3K79 methylation has similarly been implicated in gene repression in other eukaryotic lineages[7], underscoring a capacity for the DOT1 pathway to be independently recruited to regulate transcriptional repression during eukaryotic evolution.

By investigating chromatin landscapes across diverse brown algal species, we established a defined set of combinatorial chromatin states (signatures) that are highly predictive of gene expression across the clade. Active transcription was consistently associated with H3K4me3, H3K9ac and H3K36me3, whereas H3K79me2 and H4K20me3 were associated with reduced transcription. The localization of H3K79me2 varied among the brown algal clades, and its association with reduced transcription could reflect either a direct repressive role or a secondary consequence of low or no transcriptional activity, highlighting the need for future functional studies to characterize this pathway. In addition to transcriptional states, chromatin signatures were also strongly correlated with the evolutionary age of genes and the breadth of their expression. Notably, repressive chromatin signatures were highly enriched among young species-specific genes with restricted gene expression patterns, a pattern mirrored across a wide range of eukaryotes[42–44]. This supports the idea that heterochromatic regions may serve as a cradle for the emergence of novel genes[22], with silencing followed by eventual reprogramming and expression in reproductive cell types providing a route for the evolution and selection of new gene functions[44–46].

Our findings have also shed light on the regulation of sex determination and sex chromosome evolution in brown algae. Their UV sex chromosomes are enriched for repressive chromatin and have reduced conservation of chromatin signatures compared with autosomes, consistent with suppressed recombination and gene turnover in the SDRs[15,47–49]. Similar features have been reported in plants, where the *Marchantia* V chromosome is enriched for heterochromatic marks such as H3K9me1 and H3K27me3[50], suggesting that repressive chromatin may be a shared hallmark of UV sex chromosomes across distantly related lineages. Comparative studies suggest that sex-biased regulatory evolution is often lineage- and sex-specific. In plants, the evolution of dioecy is frequently associated with a disproportionate accumulation of female-biased regulatory changes[51], while in brown algae transitions between dioicy and co-sexuality are accompanied by adaptive gene expression shifts and rapid sequence evolution, particularly for male-biased genes, with co-sexual species often resembling

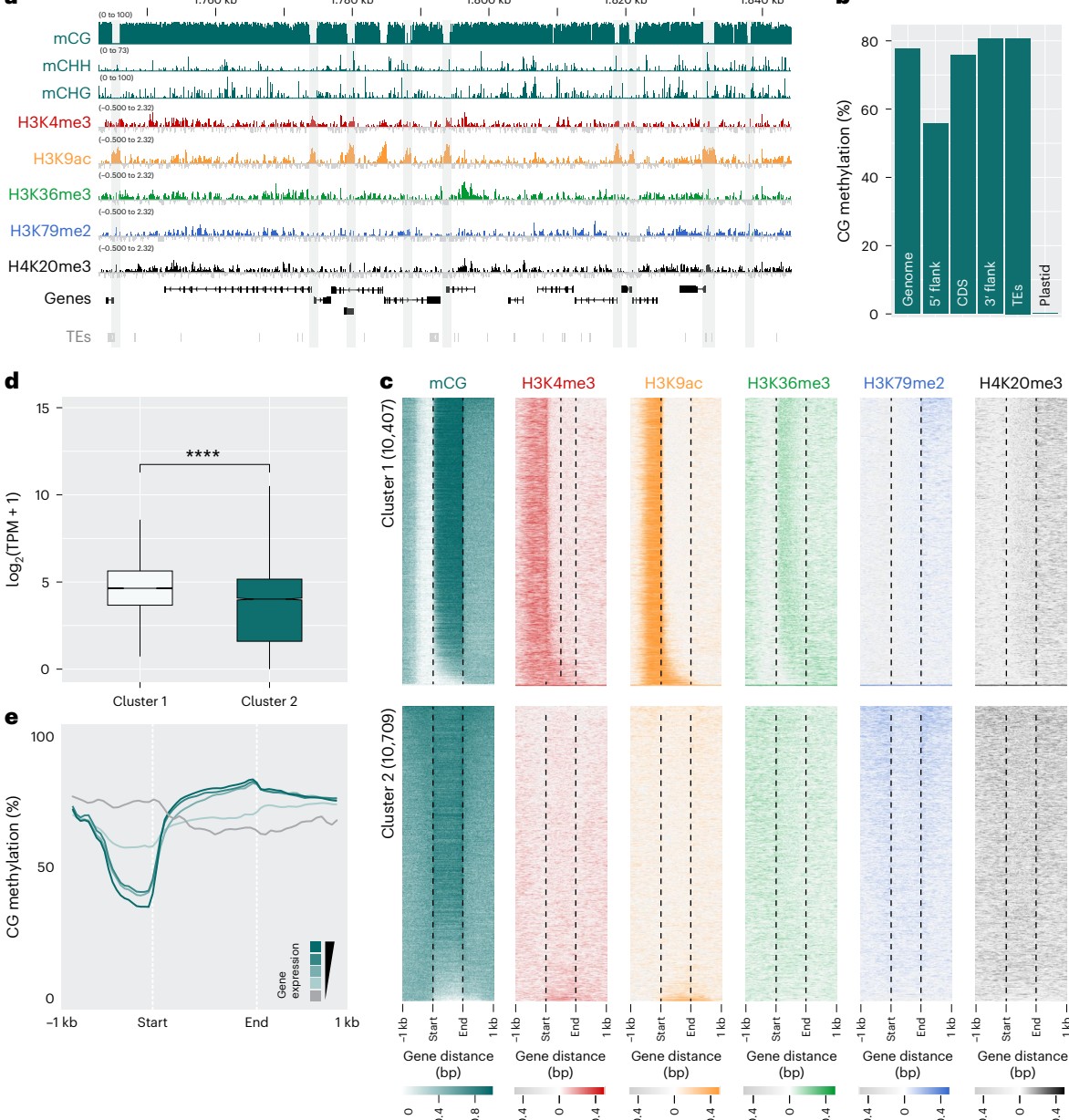

**Fig. 5 | Epigenetic landscape in the closest outgroup of the brown algae.**
**a**, Genome browser view of ChIP-seq and CG methylation tracks in *Schizocladia ischiensis*. Demethylated CpG-like islands are highlighted with grey shading, with tracks for genes and repeats shown below. ChIP-seq signals represent the log2 of enrichment of ChIP DNA relative to H3, while the CG tracks represent percentage methylation. **b**, Average genome-wide CG methylation levels at different genomic features of the nuclear genome and plastid genome. **c**, The chromatin and CG methylation landscape of *S. ischiensis* genes clustered based on the presence (cluster 1) and absence (cluster 2) of promoter demethylation. **d**, Gene expression

level of genes forming part of cluster 1 and cluster 2 shown in **c**. The boxplot shows medians (centre line), interquartile ranges (boxes) and whiskers extending to the most extreme values within 1.5× the interquartile range. *P* values were computed via the unpaired two-tailed Wilcoxon test. ****$P < 0.001$. **e**, Metaplot of 5mCG levels over genes grouped into equal quantiles based on high to low gene expression in each sample (dark turquoise to light turquoise). Genes with no expression (TPM = 0) are also shown (grey). Gene bodies are plotted as proportional lengths, upstream and downstream intergenic regions in kilobases. TE, transposable element; CDS, protein-coding sequence.

the female program[52]. With this in mind, the species-specific chromatin reconfigurations we report, which ranged from changes at PAR and sex-biased genes in species with weak sexual dimorphism to consistent enrichment among female-biased loci in other lineages, support a model in which the evolution of sexual differentiation systems proceeds through lineage-specific, modular regulatory changes rather than a common mechanism. Interestingly, even after the transition from dioicy to co-sexuality in the Desmarestiales, the former sex chromosome retains its distinct heterochromatic profile, indicating that epigenomic features associated with sex chromosomes persist

long after recombination resumes. Chromatin modifications thus not only reflect current transcriptional states but may also shape the long-term evolutionary trajectory of the sex homologue during transitions towards co-sexuality by constraining or biasing subsequent regulatory evolution. This echoes findings from plants and animals, where epigenetic silencing marks, dosage-compensation mechanisms and heterochromatin expansion continue to influence genome function long after sex chromosome turnovers or fusions. Such epigenomic legacies may therefore represent a general principle that shapes the long-term evolutionary trajectory of sex determination systems.

In summary, our findings highlight evolutionary innovations in the chromatin toolkit that accompanied the emergence of complex multicellularity in brown algae, where the loss of canonical repression pathways and the rise of DOT1/H3K79 methylation established a novel regulatory system that now underpins development and reproduction in this vital and unique eukaryotic lineage.

## Methods

### Genome screening and orthology inference

We initially performed genome screening to identify the components of epigenetic regulation in brown algae using blastp[53] (default parameters) by leveraging recently published genomic data of species in the brown algal lineage (Phaeophyceae) and early diverging Ochrophytina[12,15] along with representatives from across major eukaryotic groups. We selected five species covering the brown algal phylogenetic diversity and one outgroup species. Orthology inference was performed on these six species using OrthoFinder v2.5.5[54].

Gene trees for DOT1 subfamilies (DOT1.1, DOT1.2, DOT1.3, DOT1.4 and DOT1.5, inferred via OrthoFinder) were reconstructed by applying mafft-einsi v7.526 for multiple sequence alignment[55], ClipKIT v2.7.0 (`kpic-smart-gap` mode) for removal of gaps and uninformative sites[56], and IQ-TREE v3.0.1 (`-m MFP -B 1000 -bnni -alrt 1000 -keep-ident`) for maximum-likelihood phylogenetic inference[57,58]. Gene domain architectures were visualized using gggenes (github.com/wilkox/gggenes/) and protein domain predictions (from Pfam, Coils and ProSitePatterns databases) using InterProScan v5.74-105.0[59]. Expression profiles of DOT1 genes across sexes and developmental stages in *Ectocarpus* were obtained from a recently published RNA-seq dataset[60] (same data processing as described therein). Replicates were summarized by mean transcripts per million (TPM) and visualized using pheatmap v1.0.13 (https://github.com/raivokolde/pheatmap). We used the same gene tree reconstruction strategy for C-5 cytosine-specific DNA methylase genes (PFAM PF00145). For PF00145-containing genes, only a single representative sequence was retained in cases where multiple genes from the same species clustered together (at >95% sequence identity and mutual coverage), in order to minimize potential genome assembly artefacts. Clusters were identified using DIAMOND DeepClust[61]. Gene trees were visualized using GGTREE v4.1.1[62].

### Biological material

Gametophytes of the five brown algal species and the outgroup *S. ischiensis* were cultivated in 90-cm Petri dishes (Corning Gosselin, BH90B-102) containing at least 10 individuals, with Provasoli enriched seawater as described in refs. 19,63. Fertile individuals were harvested with a 70-µm strainer, then rinsed with seawater and dried with a paper towel for further processing. Light and temperature conditions were optimized for fertility, as described in Supplementary Table 1.

### hPTM profiling

**Histone extraction.** Histones were extracted from 0.5 g of frozen algae, pulverized in liquid nitrogen. The powder was then homogenized in 40 ml of M1 buffer (10 mM Na phosphate buffer pH 7, 100 mM NaCl, 1,000 mM hexylene glycol, 10 mM b-mercaptoethanol, 1x cOmplete protease inhibitor cocktail (Roche)). After filtering through two layers of Miracloth (Milipore, 475855), each sample was centrifugated at 2,000 × *g* for 10 min at 4 °C. The pellet was carefully resuspended in 80 ml M2 buffer (10 mM Na phosphate buffer pH 7, 100 mM NaCl, 10 mM MgCl$_2$, 1,000 mM hexylene glycol, 0.1% Triton X-100, 10 mM b-mercaptoethanol, 1x cOmplete EDTA-free protease inhibitor cocktail (Roche, CO-RO)) twice. This pellet was then incubated with extraction buffer (1,000 mM CaCl$_2$, 20 mM Tris-HCl pH 7.25, 1x cOmplete cOmplete, EDTA-free protease inhibitor cocktail) for 10 min on ice. Thereafter, 0.3 N of 37% HCl was added, followed by centrifugation at 10,000 × *g* for 5 min at 4 °C. The resulting supernatant was collected in a fresh Protein LoBind Tubes Eppendorf tube. Histones were precipitated with 20% tricholoacetic acid (TCA) and incubated for 10 min before 13,000 × *g* centrifugation at 4 °C for 30 min, followed by successive washes with 20% ice-cold TCA, ice-cold acetone supplemented with 0.2% HCl and ice-cold acetone. The pellet was dried at room temperature and resuspended in Milli-Q water overnight at 4 °C.

**Western blot.** Histone samples were supplemented with Laemmli 2x and 100 mM of DTT and NaOC until blue colouration was observed and incubated at 95 °C for 5 min. hPTMs were detected on a 15% handcast SDS-PAGE gel, using the same antibodies listed below as in the ChIP experiment. For H3, 3–15 µg of tissue-equivalent sample was loaded onto the gel. Histone samples, corresponding to 3–15 µg of tissue equivalent for H3, were loaded. For H3K9me3, around 10 ug of tissue-equivalent sample twice that amount was used, and for other histone marks, approximately 40 ug of tissue-equivalent sample four times or more. Proteins were transferred onto a 0.45-µm nitrocellulose membrane (0.45 µm, BioRad, 1620113) on a Trans-Blot Turbo Transfer System (BioRad, 1704150). Membranes were blocked in 5% milk in 1× PBS-T for 30 min. Primary antibodies were diluted in 5% milk in 1× PBS-T and incubated for 1 hour at room temperature. These included rabbit anti-H3 (Histone H3 (D2B12) XP Rabbit mAb (ChIP Formulated), Cell Signaling Technology, 4620S), anti-H3K4me3 (Tri-Methyl-Histone H3 (Lys4) (C42D8) Rabbit mAb, CST, 9751S), anti-H3K9ac (Acetyl-Histone H3 (Lys9) (C5B11) Rabbit mAb, CST, 9649S), anti-H3K79me2 (Di-Methyl-Histone H3 (Lys79) (D15E8) XP Rabbit mAb, CST, 5427S), and anti-H4K20me3 (Tri-Methyl-Histone H4 (Lys20) (D84D2) Rabbit mAb, CST, 5737S) at 1:1,000 dilution, and anti-H3K36me3 (AbcamTri-Methyl-Histone H3 (Lys36) (D5A7) XP Rabbit mAb, Abcam, 4909S) at 1 µg ml$^{-1}$. Membranes were incubated with HRP-conjugated anti-rabbit secondary antibody (1:2,000, CST, 7074S). After further washes, membranes were developed using a 1:1 mix of Trans-Blot Turbo Transfer System (BioRad, 1704150). Images were captured with ChemiDoc Imaging System from BioRad.

**Mass spectrometry.** Samples for liquid chromatography–tandem mass spectrometry (LC–MS/MS) were prepared by migrating the extracted histone on a 14% SDS-polyacryamide gel at 100 V for 10 min. The gel was dyed with LabSafe GEL Blue (G-BIOSCIENCES, 786-35) following the manufacturer's instructions. This was followed by LC–MS/MS, performed by coupling a Vanquish Neo LC system (Thermo Scientific) to an Orbitrap Astral mass spectrometer, interfaced by a Nanospray Flex ion source (Thermo Scientific). In a subsequent round of analyses, a RSLCnano system (Ultimate 3000, Thermo Scientific) connected to an Orbitrap Exploris 480 mass spectrometer (Thermo Scientific) was additionally employed.

On the Vanquish Neo LC system, peptides were injected onto a C18 column (75-µm inner diameter × 50-cm double nanoViper PepMap Neo, 2 µm, 100 Å, Thermo Scientific) regulated also at 50 °C, and separated with a linear gradient from 100% buffer A' (H$_2$O in 0.1% formic acid) to 28% buffer B (100% CH$_3$CN in 0.1% formic acid) at a flow rate of 300 nl min$^{-1}$ over 104 min. The Orbitrap Astral mass spectrometer was run in data dependent acquisition mode and MS full scans were performed in the ultrahigh-field Orbitrap mass analyser in mass-to-charge ratio (*m/z*) ranges 380–1,200 (resolution of 240,000 at *m/z* 200; maximum injection time 100 ms; auto gain control (AGC) 300%). For the Astral MS/MS spectra, the top *N* most intense ions were isolated and subjected to further fragmentation via high-energy collision dissociation (HCD) activation with the AGC target set to 100%. We selected ions with charge state from 2+ to 6+ for screening. Normalized collision energy (NCE) was set at 30 and the dynamic exclusion at 20 s.

On the RSLCnano system, peptides were trapped on a C18 column (75-µm inner diameter × 2 cm; nanoViper Acclaim PepMapTM 100, Thermo Scientific) with buffer A (2/98 CH$_3$CN/H$_2$O in 0.1% formic acid) at a flow rate of 2.5 µl min$^{-1}$ over 4 min. Separation was performed on a 50-cm × 75-µm C18 column (nanoViper Acclaim PepMapTM RSLC, 2 µm,

100 Å, Thermo Scientific) regulated to a temperature of 50 °C with a linear gradient of 2% to 30% buffer B (100% $CH_3CN$ in 0.1% formic acid) at a flow rate of 300 nl min⁻¹ over 91 min. On the Orbitrap Exploris 480 mass spectrometer, full scans were performed in ranges $m/z$ 375–1,500 (resolution of 120,000 at $m/z$ 200; maximum injection time 25 ms; AGC 300%) and the top 20 most intense ions were isolated and subjected to further fragmentation via HCD activation at resolution of 15,000 with the AGC target set also to 100%. We also selected ions with charge state from 2+ to 6+. NCE was set at 30 and with a dynamic exclusion of 10 s.

The resulting LC–MS/MS data were searched against the species-specific histone sequences using Mascot[64]. Enzyme specificity was set to trypsin and a maximum of five missed cleavage sites were allowed. Oxidized methionine, carbamidomethylated cysteine, N-terminal acetylation, acetylation, methylation (mono, di and tri) of lysine, methylation (mono and di) of arginine, methylation of glutamic acid and aspartic acid were set as variable modifications and with a maximum of nine modifications for all Mascot searches. Maximum allowed mass deviation was set to 10 ppm for monoisotopic precursor ions and 0.02 Da for MS/MS peaks. The resulting Mascot files were further processed using myProMS v.3.10.

## ChIP-seq

To map hPTMs to the genome, we performed ChIP-seq to detect the enrichment of H3K4me3, H3K9ac, H3K36me3, H3K79me2 and H4K20me3. Each sample was prepared from approximately 0.6 g of semi-dry algal tissue (~600 individuals), which was then fixed in seawater containing 1% freshly prepared formaldehyde for 10 min. The fixed sample was quenched with 400 mM glycine in 1× PBS, followed by rinsing with fresh seawater to remove residual formaldehyde. Nuclei were isolated by grinding the cross-linked tissue in liquid nitrogen and resuspending the powder in a nuclear isolation buffer containing 0.1% Triton X-100, 125 mM sorbitol, 20 mM potassium citrate, 30 mM $MgCl_2$, 5 mM EDTA, 5 mM β-mercaptoethanol, 55 mM HEPES (pH 7.5) and 1× EDTA-free protease inhibitor cocktail (Roche, CO-RO). The suspension was homogenized using a Tenbroeck Potter, filtered through Miracloth (Millipore, 475855) and centrifuged at 3,000 × $g$ for 10 min at 4 °C. The nuclear pellet was washed twice with the same buffer and once more with buffer lacking Triton X-100, conserving the centrifuge parameters. Nuclear pellets were then lysed in 1 ml of nuclear lysis buffer total (1% SDS, 10 mM EDTA, 50 mM Tris-HCl pH 8 and protease inhibitors). Chromatin was fragmented via sonication using a Covaris E220 Evolution sonicator (settings: 25% duty cycle, 75 peak power, 200 cycles per burst, 900 s duration at 4 °C) in 8 microTUBE AFA Fiber Snap-Cap tubes. Cellular debris was cleared by centrifugation at 14,000 × $g$ for 5 min at 4 °C. The resulting chromatin-containing supernatant was diluted 1:10 with ChIP dilution buffer (1% Triton X-100, 1.2 mM EDTA, 16.7 mM Tris-HCl pH 8, 167 mM NaCl and 1× EDTA-free protease inhibitor cocktail (Roche, CO-RO)). Diluted chromatin was distributed into DNA LoBind tubes (Eppendorf) and incubated overnight at 4 °C with a 1:500 (v/v) antibody on a rotator set at 10 rpm. Antibodies were sourced from Cell Signaling Technology (anti-H3: 4620; H3K4me3: 9751S; H3K9ac: 9649S; H3K79me2: D15E8; H4K20me3: 5737S) and Abcam (H3K36me3: ab9050). Immunoprecipitation was carried out using a 1:1 mixture of protein A and G Dynabeads (Thermo Fisher Scientific, 10004D and 10002D). Following binding and sequential wash steps, immune complexes were eluted in 100 µl of direct elution buffer (0.5% SDS, 5 mM EDTA, 10 mM Tris-HCl pH 8, 300 mM NaCl). Cross-link reversal was achieved by incubating samples at 65 °C overnight with intermittent shaking. DNA was purified following digestion with Proteinase K (Fisher Scientific, 11826724) and RNase A (Roche, 10109142001). DNA extraction was performed using phenol/chloroform/isoamyl alcohol (25:24:1), followed by centrifugation at 13,800 × $g$ for 15 min at 4 °C. The aqueous phase was transferred to fresh DNA low binding tubes, mixed with 1.25 ml of 100% ethanol, 50 µl of 3 M sodium acetate (pH 5.2) and 4 µl of glycogen (20 mg ml⁻¹), and incubated at −80 °C for at

least 1 hour (or overnight) for DNA precipitation. DNA was pelleted by centrifugation at 13,800 × $g$ for 15 min at 4 °C, washed with 70% ethanol and centrifuged again under the same conditions. Pellets were air-dried and resuspended in 0.1× TE buffer. Library preparation was conducted using the NEBNext Ultra II DNA Library Prep Kit (New England Biolabs, E7645S), and sequencing was carried out on the Illumina HiSeq 3000 platform, targeting 20 million of 150-bp paired-end reads per sample.

To process the ChIP-seq data, we used nf-core/chipseq v2.0.0 with default options[65]. Publicly available datasets from wild-type male and female gametophytes of *Ectocarpus*[8] were retrieved and processed using the same workflow for consistency. Biological replicates for each species were aligned to their corresponding reference genome (Supplementary Table 1). Replicates showing high correlation, as determined by Spearman's coefficient using multiBamSummary and plotCorrelation from deepTools v3.5.1[66], were merged using samtools merge for downstream analyses. Peaks were called with macs2 (default parameters). Normalized $log_2$ coverage tracks, relative to total H3, were generated using deepTools bamCompare with a bin size of 10 bp, --scaleFactorsMethod readCount and --operation log2. Using the deepTools, we computed the correlation matrices via multiBigwigSummary (in bins mode) followed by plotCorrelation. Genome-wide signal profiles were visualized in IGV v2.18.4.

## RNA-seq

RNA-seq data were generated from culture with the same conditions to match the hPTM data with the gene expression data. Each RNA-seq was carried out in triplicate. For each replicate, approximately 10 mg of algal tissue was gently blotted dry and immediately flash-frozen in liquid nitrogen. Total RNA was extracted following the method described by ref. 10. Briefly, frozen tissue was ground in liquid nitrogen and incubated at 65 °C in 700 µl of preheated CTAB3 extraction buffer (100 mM Tris-HCl pH 8, 1.4 M NaCl, 20 mM EDTA pH 8, 2% CTAB, 2% PVP and 1% β-mercaptoethanol). The lysate was vortexed and maintained at 65 °C for 5–20 min. Phase separation was achieved by extraction with chloroform:isoamyl alcohol (24:1), followed by two rounds of centrifugation at 10,000 × $g$ for 15 min at 4 °C. RNA was precipitated overnight at −20 °C using 3 M LiCl and 1% β-mercaptoethanol, pelleted by centrifugation at 10,000 × $g$ for 1 hour at 4 °C, washed with cold 70% ethanol and resuspended in RNase-free water. Genomic DNA contamination was removed using the TURBO DNase Kit (Thermo Fisher, AM1907). RNA-seq libraries were prepared using the NEBNext Ultra II Directional RNA Library Prep Kit (New England Biolabs, E7760S) and sequenced on the Illumina Next Seq 2000 platform, generating 25–30 million 150-bp paired-end reads per sample.

Reads were processed using the nf-core/rnaseq pipeline v3.12.0[65]. Genome assembly versions are described in Supplementary Table 1. Publicly available RNA-seq datasets from *Ectocarpus* sp. 7 male and female gametophytes[8] were reprocessed using the same workflow for consistency. Reproducibility across biological replicates was confirmed via Spearman correlation. Differential expression analysis between male and female was performed with DESeq2 v1.42.1[67], identifying differentially expressed genes based on a |$log_2$-fold change| ≥1 and an adjusted $P$ value < 0.05.

## Generation of histone mark profiles

To investigate the distribution of histone modifications across genes grouped by expression level, we used a custom bash script for automated signal processing and plotting using deepTools v3.5.1 within a conda environment. Genes were divided into five groups: one group comprising genes with zero expression (0 TPM), and the remaining genes divided into four groups according to expression level quartiles—one where there was no expression and the rest split by quartiles (calculated for each sample). Gene coordinates grouped by expression quantiles were provided as BED files, and ChIP-seq signal was taken from precomputed bigWig tracks. Signal matrices were generated

using computeMatrix scale-regions -a 1000 -b 1000 -bs 100 −skipZeros, which summarized ChIP-seq signal over gene bodies (represented as 5 kb) and 1 kb flanking regions, with a bin size of 100 bp. Regions with zero signal were excluded. The resulting matrices were visualized using plotProfile --plotType se, producing average signal profiles per quantile group.

To test the repressive role of H3K79me2 at the gene level, we used the R package GeneOverlap v1.40.0 (https://github.com/shenlab-sinai/GeneOverlap), performing Fisher's exact tests and visualizing odds ratios for all pairwise overlaps between H3K79me2 signal and expression quantiles (as described above).

### Chromatin emission state and signature inference

**Detection of emission states with hiHMM.** The five chromatin marks were analysed using hiHMM[21] to annotate each genome with emission states. Input files for hiHMM were generated in several steps following the recommendations from ref. 21. First, BedGraph files were produced using bamCompare -bs 200 --scaleFactorsMethod readCount --pseudocount 0.5 --operation log2 -of 'bedgraph'. Then BedGraph files were normalized prior to modelling using a short Python script that applied sklearn.preprocessing.StandardScaler.fit_transform() to the signal column (the fourth). Quality checks involving visual inspection of quantile–quantile plots confirmed standardization. Normalized BedGraphs were reformatted into chromosome-wise matrices with genomic bins as rows and samples as columns, and file and chromosome names were standardized as required.

The hiHMM model was trained on a reduced subset of chromosomes (one autosome per sex, plus one male and one female sex chromosome or sex homologue when available), which reduces computational complexity without affecting model quality according to the authors[21]. For each species, we selected the two longest scaffolds together with the available sex chromosome(s). The list of training chromosomes and run parameters are provided in Supplementary Table 13. The optimized model initiates with $K_0 = 7$ and ends up with $K = 27$ states plus one 'E0' state containing regions where no reads were assigned. The decoding part was carried on all scaffolds. The emission matrix can be found in Fig. 2b and the transition matrix can be found in Extended Data Fig. 4 as well as the input files used in Supplementary Dataset 1. Model 2 was favoured, as Model 1 did not provide an integrated or streamlined representation of the data. To examine the hiHMM model, we adapted the advanced functions of ChromHMM[68]. The optimization of hiHMM model was obtained using ChromHMM CompareModels function where the hiHMM model was tested for different value of $K_0$. From these comparisons, we decided to use $K_0 = 7$ (as the default). The overlap enrichment of genomic features was produced with ChromHMM OverlapEnrichment.

**Definition of chromatin signatures.** To streamline the analysis, emission states assigned to each gene were consolidated into five broader categories based on their predominant emission enrichment with genomic features as done in ref. 69. States showing strong enrichment at TSSs were classified under the 'TSS' category (E5, E10, E12). TSSs were chosen for this analysis because transcription start sites remain unannotated in these brown algal genomes. Emission states predominantly associated with gene bodies were grouped into the 'Genic' category (E8, E2, E7, E9, E25). States displaying mixed enrichment patterns—with or without H3K36me3—were collectively grouped under 'Mixed,' due to the absence of clear feature-specific enrichment (E14, E17, E19, E4, E24, E18, E15, E11, E6, E20, E26, E22, E23). A distinct 'Silenced' category was defined for states marked exclusively by H3K79me2 and H4K20me3 (specifically E27 and E1). Finally, states characterized by minimal or no detectable histone modification signals were assigned to the 'Null states' group (E0, E3, E13, E16, E21). Each gene was then annotated based on its combination of these five chromatin categories, resulting in a defined set of unique chromatin profiles referred to as 'chromatin signatures'. Genes were assigned to the null signature group (S1) if they were associated solely with low-signal states (E0, E3, E13, E16, E21), and no other chromatin category. In total, this classification yielded 16 distinct chromatin signatures (S1–S16) across the six analysed genomes. Signatures were then labelled following the increasing gene expression median across all species.

Chromatin signatures of genes were extracted from Supplementary Tables 6–11 and compared between samples. Enrichment of chromatin changes was examined by different category type such as gene location (autosomes, PAR, SDR, sex homologue chromosome) and expression bias (female-biased, male-biased, unbiased).

### Evolutionary analysis of chromatin signatures

We analysed the overall conservation of chromatin signatures by counting the number of one-to-one orthologues with the same chromatin signature label across all six species. To examine the significance of the observed value, we performed a permutation test by reshuffling the chromatin signature labels across orthologues for each species and recomputing the number of overlaps. The permuted values represent the null assumption of no conservation of chromatin signatures. The same procedure was followed for pairwise comparisons (against *Ectocarpus*). These analyses were enabled by rsample v1.2.1 (https://rsample.tidymodels.org). Female chromatin signature data were used for dioicous species in this and following analyses. *P* values were computed from the 10,000 permutation results by comparing the observed statistic to the empirical null distribution.

To examine the influence of sex chromosomes on the signature conservation, we calculated the percentage of observed cross-species pairwise overlaps separately for autosomes and sex chromosomes for each focal species. We only considered the chromosome type in the focal species and not the target species of the comparison due to the high evolutionary turnover rate of one-to-one orthologues in the sex chromosome.

To profile the gene-wise evolutionary signal associated with each chromatin signature, we performed genomic phylostratigraphy using GenEra[23] to infer the evolutionary age of each gene, with the resulting phylostratigraphic rank (phylorank) ranging from 1 (cellular organisms) to 15 (species level) in *Ectocarpus*. It should be noted that the phylorank of the youngest genes differs between species due to differences in the presence of genomic data at each taxonomic node. For the orphan gene analysis, gene models with the highest phylorank for each species were considered as orphan genes, for which we then obtained their prevalence (in %) in each chromatin signature. The red arrows indicate chromatin signature with high proportion of evolutionarily young or orphan genes across multiple species. The gene age and orphan gene analyses were performed on genes with TPM > 1, to exclude gene models with weak transcriptional evidence.

Alongside the evolutionary signals, we profiled the expression variability associated with each chromatin signature. Using a recently published developmental RNA-seq dataset of *Ectocarpus*[60] we calculated tau scores[70] to estimate the expression specificity of genes, which ranges from 0 (broadly expressed) to 1 (developmental stage-specific expression), using the median expression values for each gene across replicates. For other species, which lack comparably comprehensive RNA-seq datasets, we calculated expression variability using the RNA-seq data produced in this study as input. Specifically, we computed the relative entropy (using KL.empirical() from entropy v1.3.2) for each gene between the observed $\log_2(\text{TPM} + 1)$ expression values across replicates and a null assumption of uniform expression for each gene. To reduce the effect of noise in lowly expressed genes, genes with mean TPM below 3 were discarded for this analysis.

### Data analytics and graphics

Statistical analysis, data processing and visualization were done using R v4.3.3 with the following packages: tidyverse v2.0.0[71] (ggplot2 v3.5.2,

dplyr v1.1.4, tidyr v1.3.1, tibble v3.3.0, stringr v1.5.1, readxl v1.4.3, purrr v1.1.0), pheatmap v1.0.13 (https://github.com/raivokolde/pheatmap), ggbeeswarm v0.7.2 (https://github.com/eclarke/ggbeeswarm), rtracklayer v1.64.0[72], duckplyr v0.4.1 (https://duckplyr.tidyverse.org/), GenomicRanges v1.56.2, ggalluvial v0.12.5 (https://corybrunson.github.io/ggalluvial/), ggrastr v1.0.2 (https://github.com/VPetukhov/ggrastr) and rsample v1.2.1 (https://rsample.tidymodels.org). Statistical tests and significance levels are indicated in the text, figure legends and Supplementary Table 14. Scripts are provided at https://github.com/jerovign/PhaeoChromo.

### Reporting summary

Further information on research design is available in the Nature Portfolio Reporting Summary linked to this article.

### Data availability

Mass spectrometry data are available via ProteomeXchange under identifier PXD065559. ChIP-seq and RNA-seq short-read data have been uploaded to NCBI under BioProject PRJNA1328953. Supplementary Dataset 1 can be accessed at https://doi.org/10.17617/3.TDGYHS.

### Code availability

Scripts are available on GitHub (https://github.com/jerovign/PhaeoChromo) and archived on Zenodo at https://doi.org/10.5281/zenodo.18629747 (ref. 73).

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

## Acknowledgements

This study was funded by the Max Planck Society, the European Research Council grant 864038 (S.M.C.), the Bettencourt Foundation (S.M.C.) and the Moore Foundation (S.M.C.). The MS platform at the Curie Institut is supported by Région Ile-de-France (no. EX061034), and ITMO Cancer of Aviesan and INCa are supported by funds administered by INSERM (no. 21CQ016-00). The LSMP thanks P. Poullet from the bioinformatics core facility (CUBIC) of the Institut Curie U1331 for the continuous development of myProMS. J.S.L. thanks V. Fernández Roces and H.-G. Drost for discussions and support. J.V. and J.S.L. are thankful to the IMPRS 'From molecules to organisms'.

## Author contributions

S.M.C. and M.B. conceived and designed the experiments, and carried out project administration. J.V. performed experiments and the algae culture was supported by R.L. J.V., J.S.L., M.B. and F.B.H. analysed the data. P.L. provided unpublished data and performed experiments. B.L. carried out the mass spectrometry experimental work. D.L. supervised mass spectrometry data analysis. S.M.C. provided resources. S.M.C. and M.B. wrote the paper with input from J.V. and J.S.L. All authors read and approved the paper.

## Funding

## Competing interests

The authors declare no competing interests.

## Additional information

**Extended data** is available for this paper at

**Supplementary information** The online version contains supplementary
material available at https://doi.org/10.1038/s41559-026-03031-3.

**Correspondence and requests for materials** should be addressed to
Michael Borg or Susana M. Coelho.

**Peer review information** *Nature Ecology & Evolution* thanks Alex de
Mendoza and the other, anonymous, reviewer(s) for their contribution
to the peer review of this work. Peer reviewer reports are available.

**Publisher's note** Springer Nature remains neutral with regard
to jurisdictional claims in published maps and institutional
affiliations.

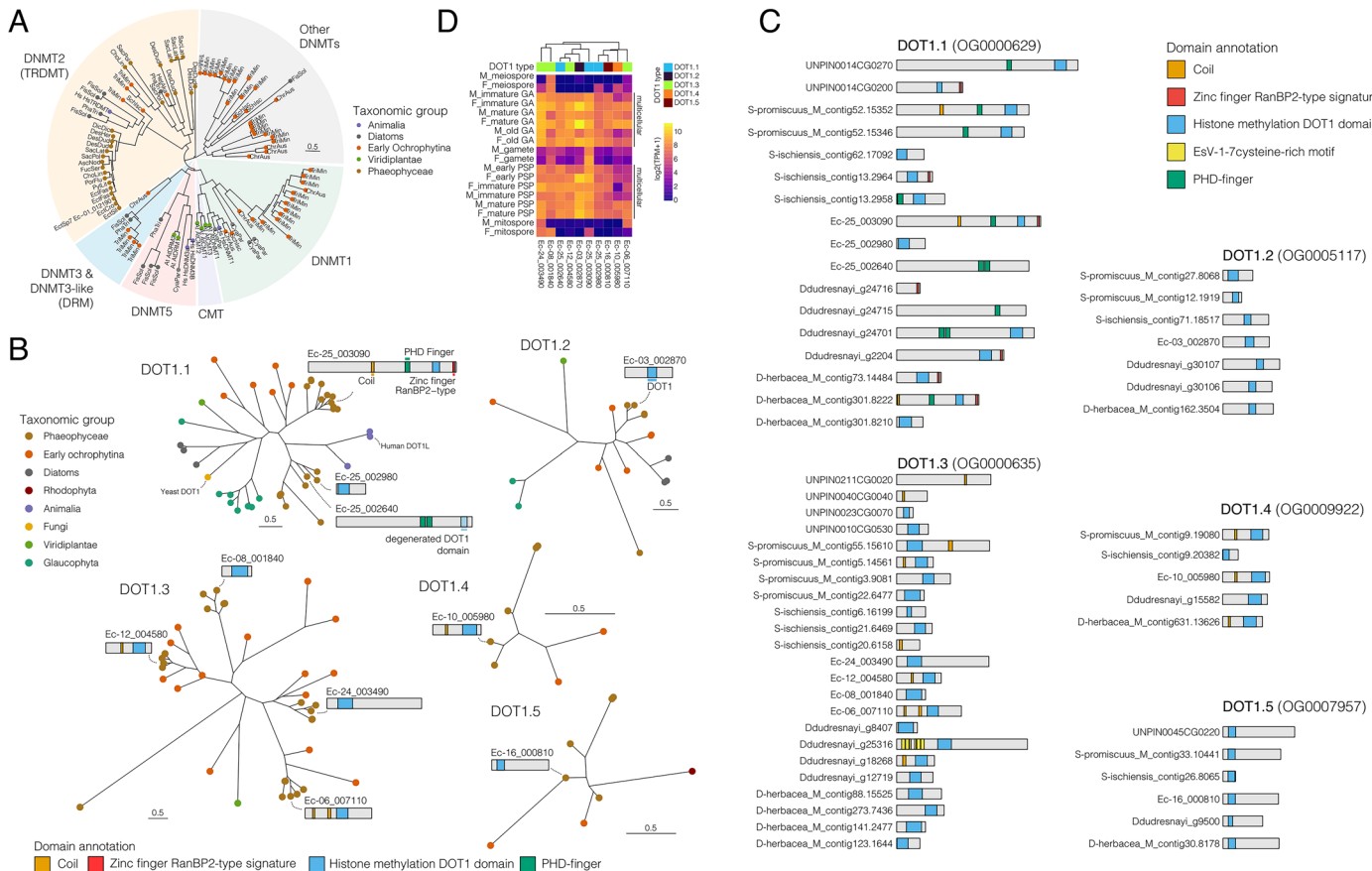

**Extended Data Fig. 1 | Evolutionary relationship of C-5 cytosine-specific DNA methylase and DOT1 genes across eukaryotes.** Related to Fig. 1a. (**A**) Maximum likelihood gene tree of C-5 cytosine-specific DNA methylase genes (PFAM PF00145). The tip colours indicate the taxonomic group. The tip labels are the first three letters of the genus and species name from Fig. 1a. Gene names are labelled for *Ectocarpus* (EctSp7), *Homo sapiens* (Hs) and *Arabidopsis thaliana* (At). (**B**) Maximum likelihood gene trees for DOT1 orthogroups (DOT1.1 to DOT1.5). The tip colours indicate the taxonomic group and the tip labels are gene name and domain architecture in *Ectocarpus*. The position of the 'degenerated

DOT1 domain', that was not detected via InterProScan, was inferred via multiple sequence alignment of *Ectocarpus* DOT1 domains (in DOT1.1) against the DOT1.1 orthogroup member Ec-25_002640. (**C**) Domain architecture of gene models from (**B**) across brown algae and the outgroup *Schizocladia ischiensis*, grouped by DOT orthogroups. (**D**) Expression profile of DOT1 genes across life cycle stages in both male and female *Ectocarpus*. Sex is denoted by prefix (M = male, F = female). Multicellular gametophytic and partheno-sporophytic generations are denoted by GA and PSP, respectively.

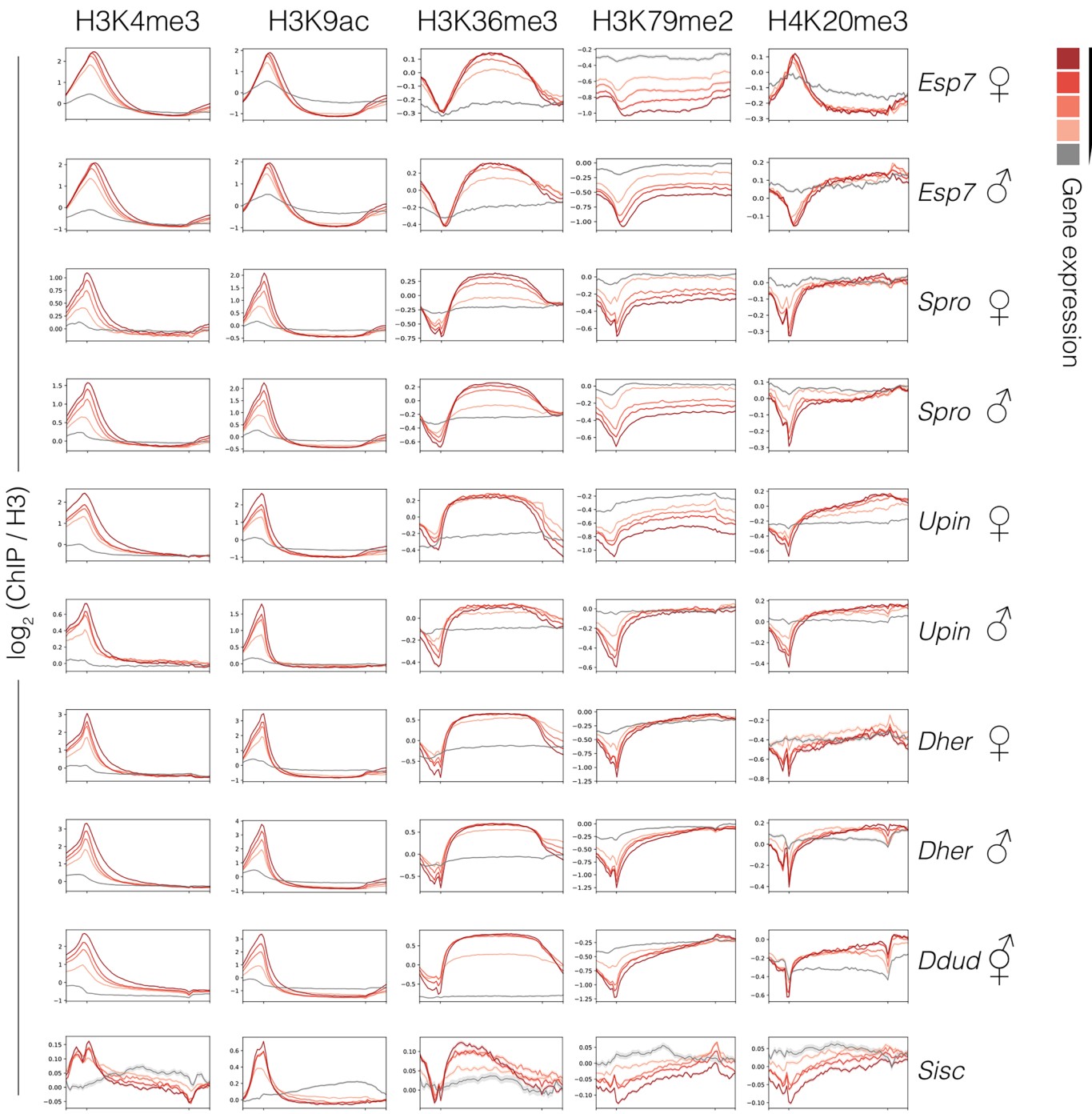

**Extended Data Fig. 2 | Metaplots of H3K4me3, H3K9ac, H3K36me3, H3K79me2 and H4K20me3 coverage over genes grouped by expression levels.** Genes were grouped into equal quantiles by high to low gene expression levels in each sample (dark red to light red). Genes with no expression (TPM = 0) are also shown (grey). Gene bodies are plotted as proportional lengths, upstream and downstream intergenic regions in kilobases. Unlike in Fig. 2e, which displays only genes overlapping an H3K79me2 peak, these metaplots include all genes across the genome of each species.

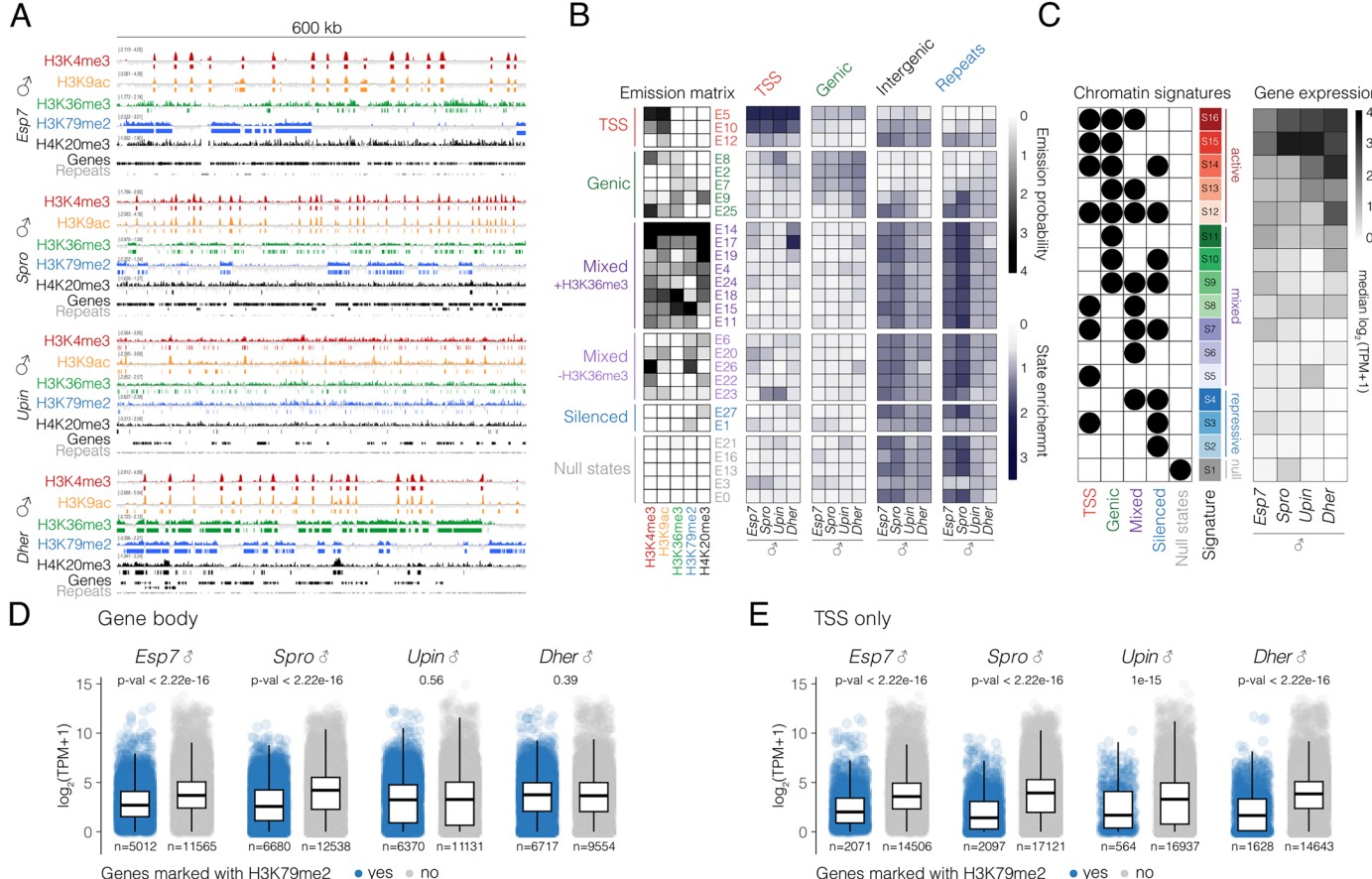

**Extended Data Fig. 3 | The chromatin landscape of brown algal male gametophytes.** (**A**) Representative genome browser tracks of ChIP-seq datasets in male datasets of the dioicous species. ChIP-seq coverage is represented as the log2 ratio of IP DNA relative to histone H3, with the range indicated on each track. ChIP-seq peaks for each hPTM are indicated under their respective track. (**B**) Enrichment of each chromatin state in genomic features in the male samples of the four dioicous species. (**C**) Median RNA-seq expression level (log2 TPM + 1) of genes associated with each signature in the male samples of the four dioicous species. (**D-E**) Gene expression levels in males for genes with (blue) or without (grey) an H3K79me2 peak overlapping their gene body (**D**) or TSS (**E**). P-values were computed via the unpaired two-tailed Wilcoxon test in (**D-E**).

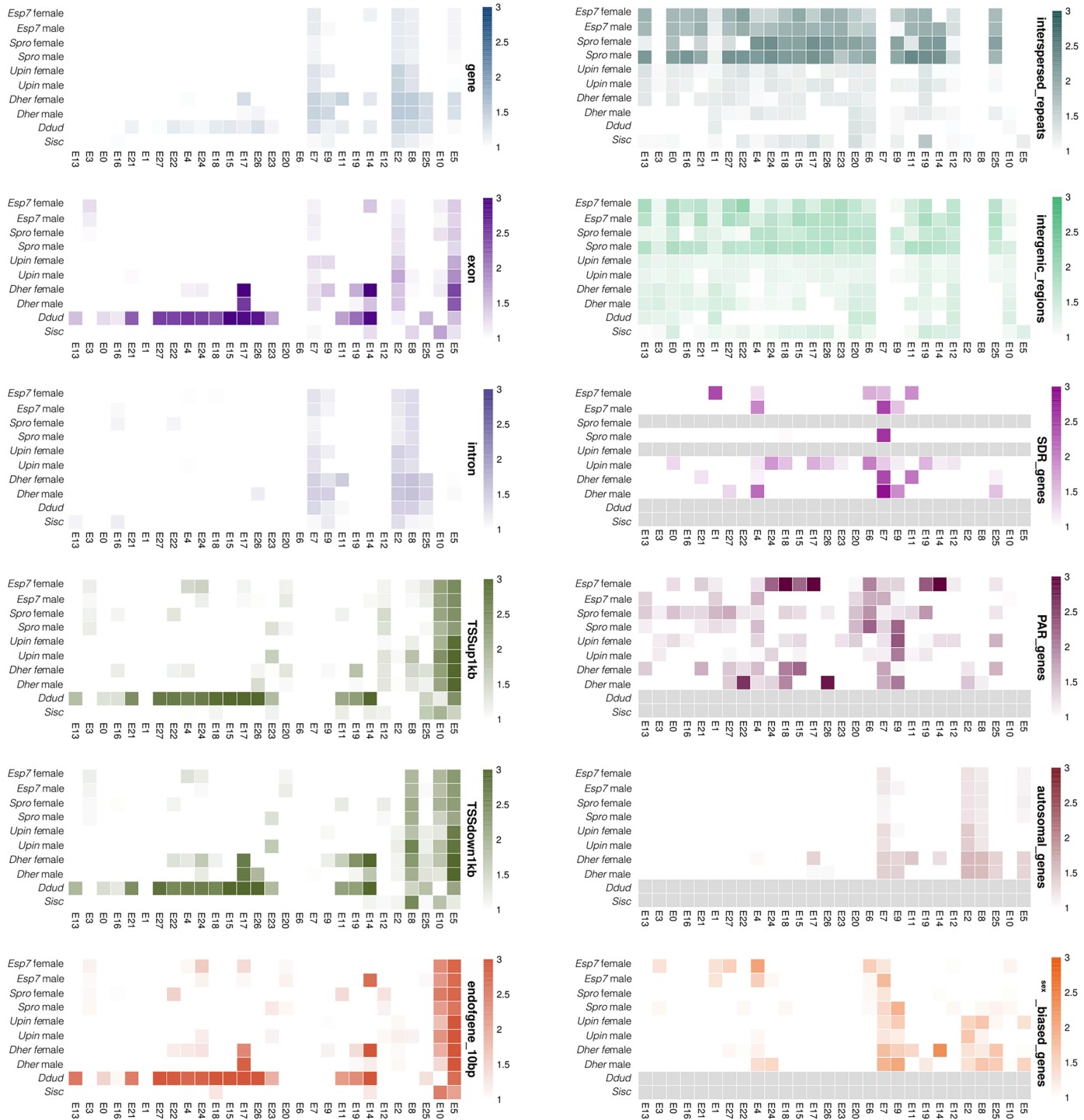

**Extended Data Fig. 4 | Enrichment of hiHMM emission states over genomic features across species.** SDR: sex-determining region; PAR: pseudo-autosomal region.

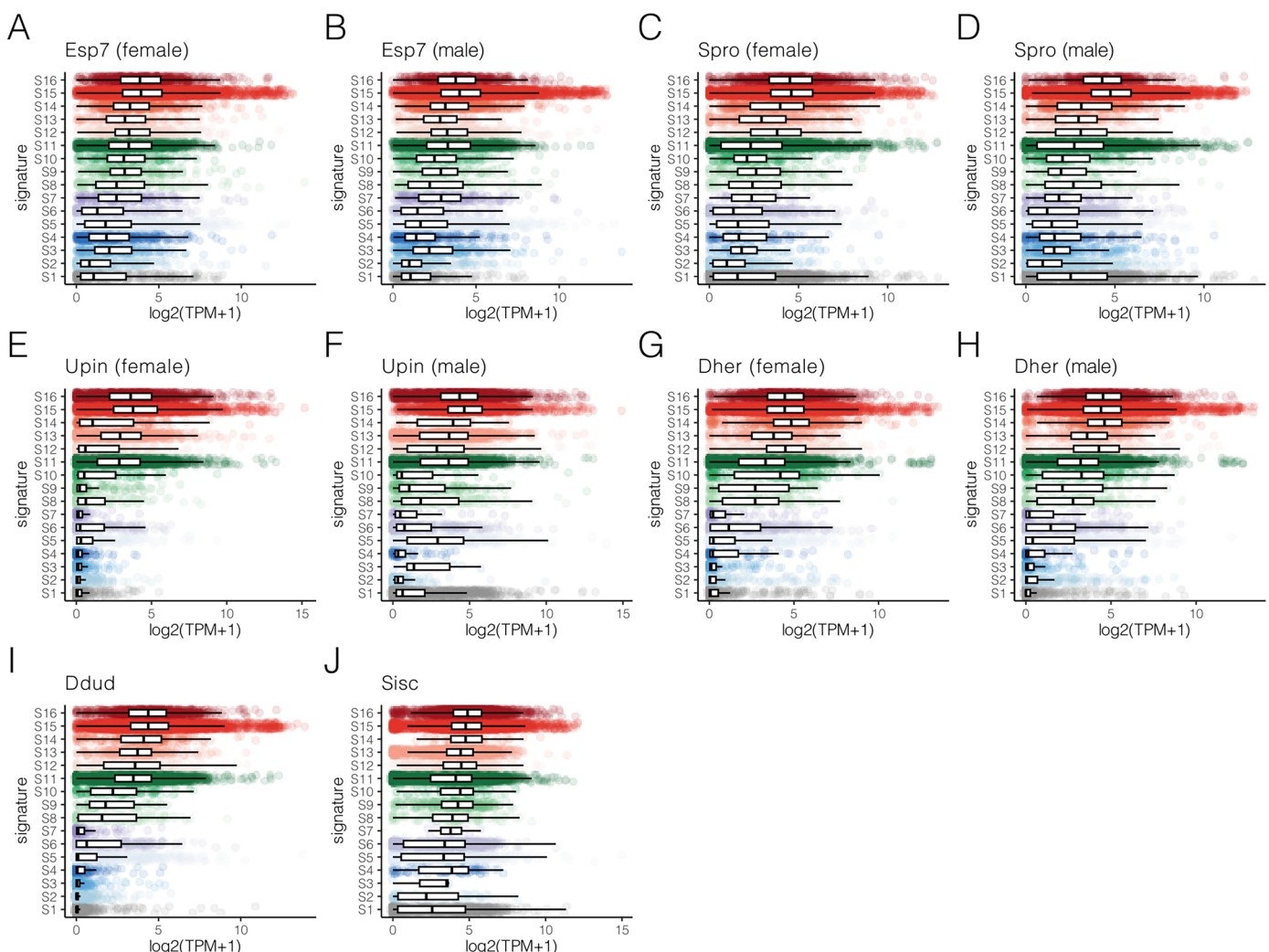

**Extended Data Fig. 5 | Distribution of gene expression level across each chromatin signature.** Gene expression level is quantified as the mean TPM across replicates, followed by log2 transformation. For dioicous species, ChIP-seq signatures and RNA-seq data are derived from samples of the same sex. (**A**) *Ectocarpus* sp.7 female, (**B**) *Ectocarpus* sp.7 male, (**C**) *Scytosiphon promiscuus* female, (**D**) *S. promiscuus* male, (**E**) *Undaria pinnatifida* female, (**F**) *U. pinnatifida* male, (**G**) *Desmarestia herbacea* female, (**H**) *D. herbacea* male, (**I**) *Desmarestia dudresnayi*, and (**J**) *Schizocladia ischiensis*. The number of genes analysed are as follows: 16447 in Esp7, 19152 in Spro, 11866 in Upin; 16127 in Dher, 18164 in Ddud, and 21022 in Sisc. The boxplots show medians (center line), interquartile ranges (boxes) and whiskers extending to the most extreme values within 1.5× the interquartile range.

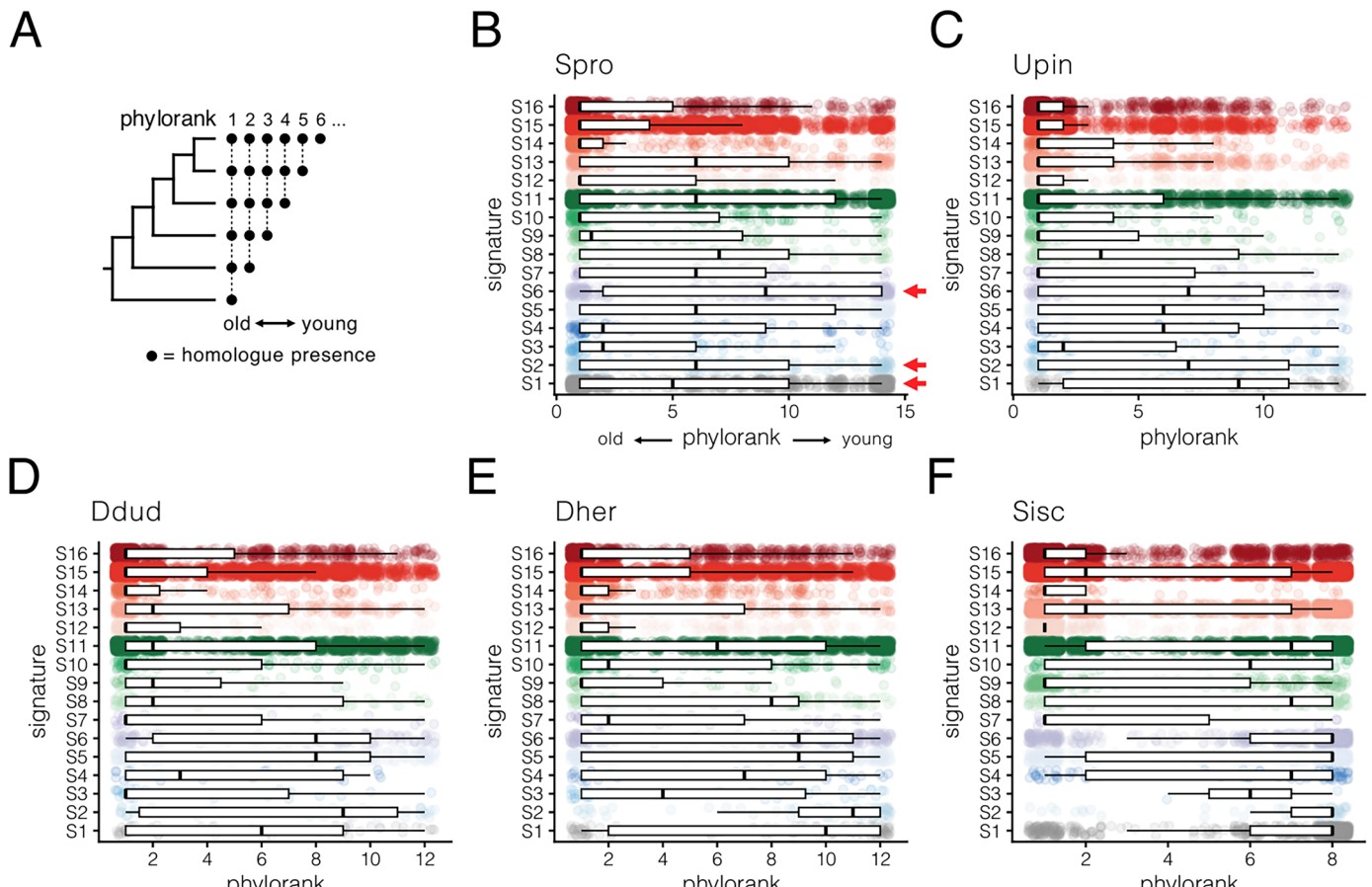

**Extended Data Fig. 6 | Distribution of gene age across each chromatin signature.** (**A**) Summary of the approach used to infer gene age (phylorank) via pairwise sequence alignment. Genes with homolog(s) detected at the deepest taxonomic node (that is cellular organisms) are assigned a phylorank of 1, whereas genes without any detectable homologs in any other species are given the highest phylorank, which corresponds to the total number of taxonomic nodes defined for the given species. A large subset of genes fall in between these two extremes. The resulting gene age distribution across chromatin signatures are shown for (**B**) *Scytosiphon promiscuus*, (**C**) *Undaria pinnatifida*,

(**D**) *Desmarestia dudresnayi*, (**E**) *Desmarestia herbacea*, and (**F**) *Schizocladia ischiensis*. The red arrows mark the chromatin signatures with consistently high distribution of young genes across species. Related to Fig. 3c. For dioicous species, female chromatin signatures were used. The number of genes analysed are as follows: 16106 in Spro, 9022 in Upin; 13547 in Dher, 13172 in Ddud, and 18519 in Sisc. The boxplots show medians (center line), interquartile ranges (boxes) and whiskers extending to the most extreme values within 1.5× the interquartile range.

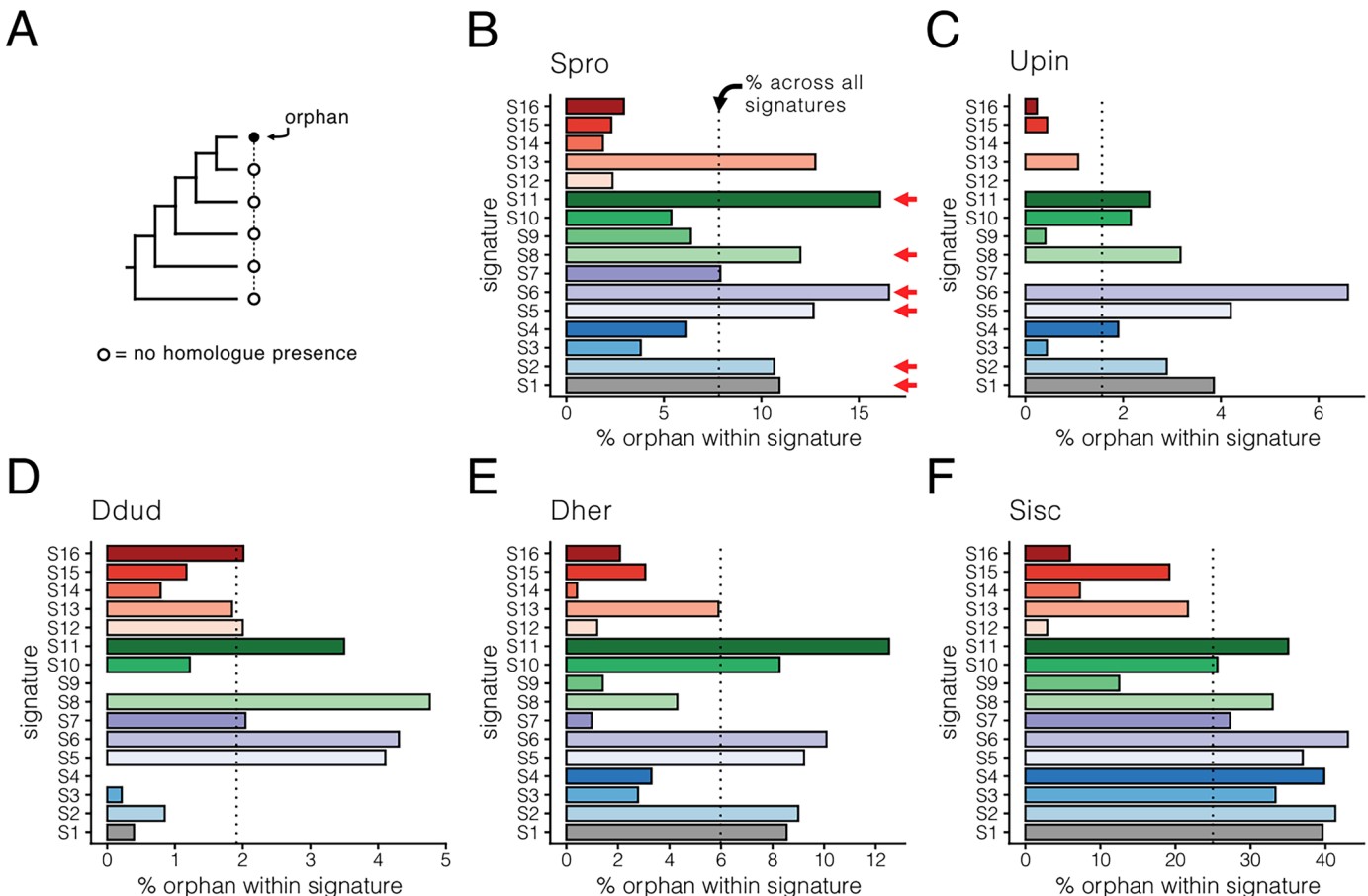

**Extended Data Fig. 7 | Percentage of orphan genes assigned to each chromatin signature.** (**A**) Summary of the approach used to infer orphan genes via pairwise sequence alignment. Genes without detectable homologs outside of the focal species is assigned as an orphan gene. The resulting percentage of orphan genes detected across chromatin signatures are shown for (**B**) *Scytosiphon promiscuus*, (**C**) *Undaria pinnatifida*, (**D**) *Desmarestia dudresnayi*, (**E**) *Desmarestia herbacea*, and (**F**) *Schizocladia ischiensis*. The red arrows mark the chromatin signatures with consistently high orphan gene presence across species. Related to Fig. 3d. For dioicous species, female chromatin signatures were used. The number of genes analysed are as follows: 16106 in Spro, 9022 in Upin; 13547 in Dher, 13172 in Ddud, and 18519 in Sisc.

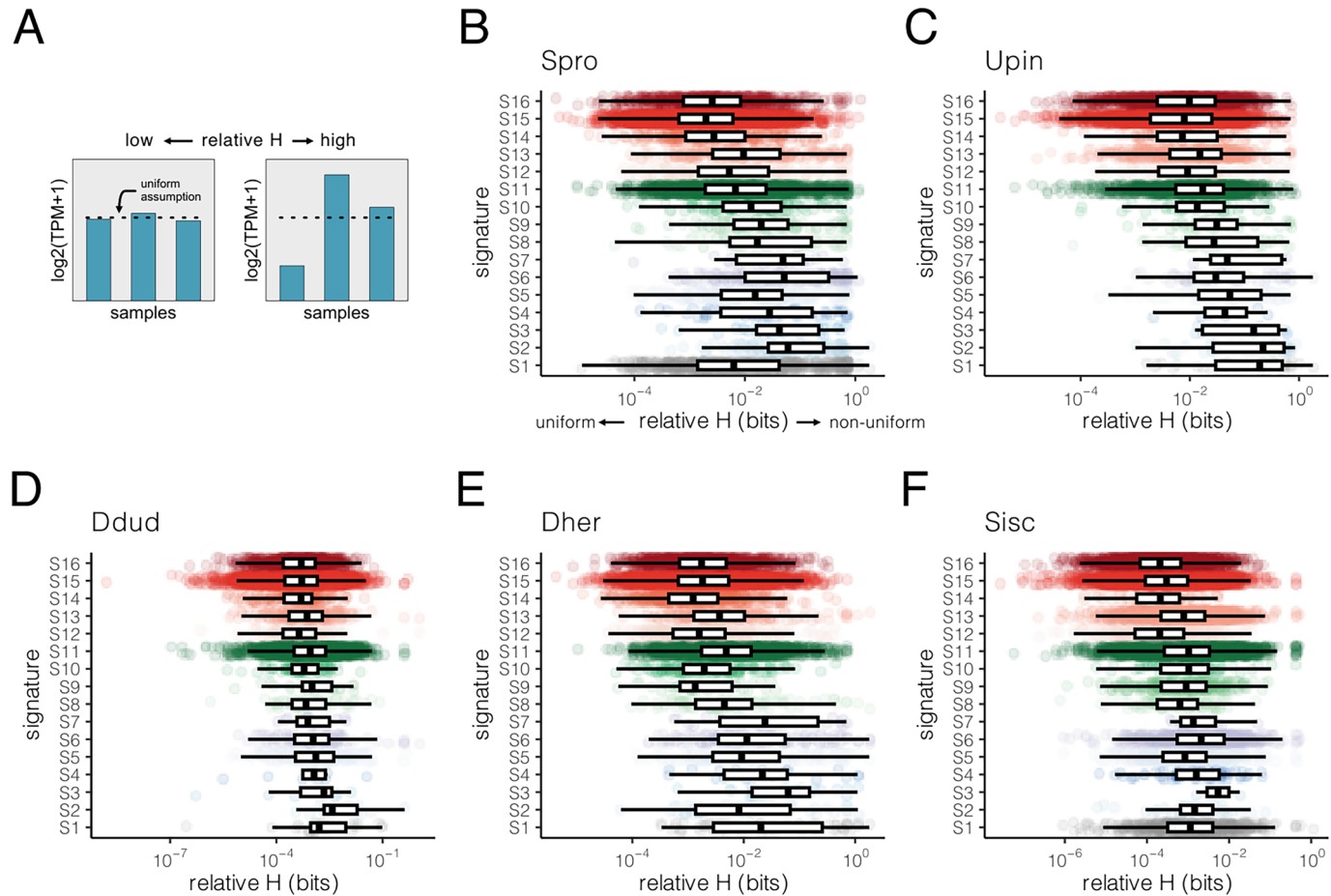

**Extended Data Fig. 8 | Distribution of expression uniformity across RNA-seq libraries for genes assigned to each chromatin signature. (A)** Summary of the relative entropy (relative H) approach to compare the observed $\log_2(TPM + 1)$ expression values across replicates and a gene-specific null assumption of uniform expression values. The resulting relative H scores across chromatin signature for **(B)** *Scytosiphon promiscuus*, **(C)** *Undaria pinnatifida*, **(D)** *Desmarestia dudresnayi*, **(E)** *Desmarestia herbacea*, and **(F)** *Schizocladia ischiensis*. Related to Fig. 3e. For dioicous species, female chromatin signatures and expression data were used. The number of genes analysed are as follows: 13582 in Spro, 7487 in Upin; 12336 in Dher, 12028 in Ddud, and 17642 in Sisc. The boxplots show medians (center line), interquartile ranges (boxes) and whiskers extending to the most extreme values within 1.5× the interquartile range.

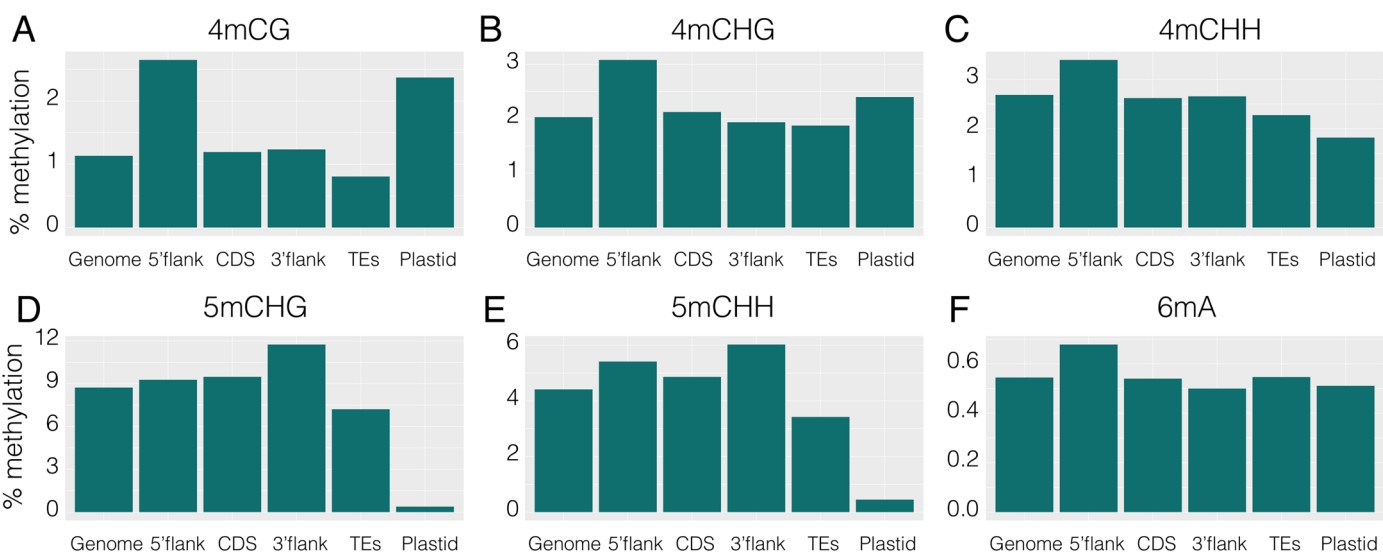

**Extended Data Fig. 9 | DNA methylation in *S. ischiensis* outside of 5mCG contexts.** Average genome-wide levels of (**A-C**) 4mC, (**D-E**) 5mC and (**F**) 6 mA in all relevant contexts at different genomic features of the nuclear genome and plastid genome.

# Reporting Summary

## Statistics

For all statistical analyses, confirm that the following items are present in the figure legend, table legend, main text, or Methods section.

| n/a | Confirmed | |
|---|---|---|
| ☐ | ☒ | The exact sample size (*n*) for each experimental group/condition, given as a discrete number and unit of measurement |
| ☐ | ☐ | A statement on whether measurements were taken from distinct samples or whether the same sample was measured repeatedly |
| ☐ | ☒ | The statistical test(s) used AND whether they are one- or two-sided <br> *Only common tests should be described solely by name; describe more complex techniques in the Methods section.* |
| ☒ | ☐ | A description of all covariates tested |
| ☐ | ☒ | A description of any assumptions or corrections, such as tests of normality and adjustment for multiple comparisons |
| ☒ | ☐ | A full description of the statistical parameters including central tendency (e.g. means) or other basic estimates (e.g. regression coefficient) AND variation (e.g. standard deviation) or associated estimates of uncertainty (e.g. confidence intervals) |
| ☐ | ☒ | For null hypothesis testing, the test statistic (e.g. *F*, *t*, *r*) with confidence intervals, effect sizes, degrees of freedom and *P* value noted <br> *Give P values as exact values whenever suitable.* |
| ☒ | ☐ | For Bayesian analysis, information on the choice of priors and Markov chain Monte Carlo settings |
| ☒ | ☐ | For hierarchical and complex designs, identification of the appropriate level for tests and full reporting of outcomes |
| ☐ | ☒ | Estimates of effect sizes (e.g. Cohen's *d*, Pearson's *r*), indicating how they were calculated |

*Our web collection on statistics for biologists contains articles on many of the points above.*

## Software and code

Policy information about availability of computer code

| | |
|---|---|
| Data collection | *Provide a description of all commercial, open source and custom code used to collect the data in this study, specifying the version used OR state that no software was used.* |
| Data analysis | All custom code and software versions are detailed here: https://github.com/jerovign/PhaeoChromo (DOI: 10.5281/zenodo.18629748), as well as in DOI 10.17617/3.TDGYHS and in the MS. |

For manuscripts utilizing custom algorithms or software that are central to the research but not yet described in published literature, software must be made available to editors and reviewers. We strongly encourage code deposition in a community repository (e.g. GitHub). See the Nature Portfolio guidelines for submitting code & software for further information.

## Data

Policy information about availability of data

All manuscripts must include a data availability statement. This statement should provide the following information, where applicable:
- Accession codes, unique identifiers, or web links for publicly available datasets
- A description of any restrictions on data availability
- For clinical datasets or third party data, please ensure that the statement adheres to our policy

Mass spectrometry data are available via ProteomeXchange under identifier PXD065559. ChIP-seq and RNA-seq short-read data has been uploaded to NCBI under

BioProject PRJNA1328953. Supplemental Dataset 1 can be accessed at https://doi.org/10.17617/3.TDGYHS. Link for all datasets for reviewers can be found at https://dataview.ncbi.nlm.nih.gov/object/PRJNA1328953?reviewer=64olmhkp43fhmu07q3t2g4iddj

## Research involving human participants, their data, or biological material

Policy information about studies with human participants or human data. See also policy information about sex, gender (identity/presentation), and sexual orientation and race, ethnicity and racism.

| | |
|---|---|
| Reporting on sex and gender | *Use the terms sex (biological attribute) and gender (shaped by social and cultural circumstances) carefully in order to avoid confusing both terms. Indicate if findings apply to only one sex or gender; describe whether sex and gender were considered in study design; whether sex and/or gender was determined based on self-reporting or assigned and methods used. Provide in the source data disaggregated sex and gender data, where this information has been collected, and if consent has been obtained for sharing of individual-level data; provide overall numbers in this Reporting Summary. Please state if this information has not been collected. Report sex- and gender-based analyses where performed, justify reasons for lack of sex- and gender-based analysis.* |
| Reporting on race, ethnicity, or other socially relevant groupings | *Please specify the socially constructed or socially relevant categorization variable(s) used in your manuscript and explain why they were used. Please note that such variables should not be used as proxies for other socially constructed/relevant variables (for example, race or ethnicity should not be used as a proxy for socioeconomic status). Provide clear definitions of the relevant terms used, how they were provided (by the participants/respondents, the researchers, or third parties), and the method(s) used to classify people into the different categories (e.g. self-report, census or administrative data, social media data, etc.) Please provide details about how you controlled for confounding variables in your analyses.* |
| Population characteristics | *Describe the covariate-relevant population characteristics of the human research participants (e.g. age, genotypic information, past and current diagnosis and treatment categories). If you filled out the behavioural & social sciences study design questions and have nothing to add here, write "See above."* |
| Recruitment | *Describe how participants were recruited. Outline any potential self-selection bias or other biases that may be present and how these are likely to impact results.* |
| Ethics oversight | *Identify the organization(s) that approved the study protocol.* |

Note that full information on the approval of the study protocol must also be provided in the manuscript.

# Field-specific reporting

Please select the one below that is the best fit for your research. If you are not sure, read the appropriate sections before making your selection.

☒ Life sciences ☐ Behavioural & social sciences ☐ Ecological, evolutionary & environmental sciences

For a reference copy of the document with all sections, see nature.com/documents/nr-reporting-summary-flat.pdf

# Life sciences study design

All studies must disclose on these points even when the disclosure is negative.

| | |
|---|---|
| Sample size | several hundred individuals per species and sex were used |
| Data exclusions | no data was excluded |
| Replication | all experiments were replicated |
| Randomization | petri dishes with growing algae were positioned randomly in the culture room to ensure homogeneous growth conditions |
| Blinding | n.a |

# Reporting for specific materials, systems and methods

We require information from authors about some types of materials, experimental systems and methods used in many studies. Here, indicate whether each material, system or method listed is relevant to your study. If you are not sure if a list item applies to your research, read the appropriate section before selecting a response.

## Materials & experimental systems

| n/a | Involved in the study |
|---|---|
| ☐ | ☒ Antibodies |
| ☒ | ☐ Eukaryotic cell lines |
| ☒ | ☐ Palaeontology and archaeology |
| ☒ | ☐ Animals and other organisms |
| ☒ | ☐ Clinical data |
| ☒ | ☐ Dual use research of concern |
| ☒ | ☐ Plants |

## Methods

| n/a | Involved in the study |
|---|---|
| ☐ | ☒ ChIP-seq |
| ☒ | ☐ Flow cytometry |
| ☒ | ☐ MRI-based neuroimaging |

# Antibodies

| | |
|---|---|
| Antibodies used | details are provided in tables and methods in the main manuscript |
| Validation | details provided in the manuscript |

# Plants

| | |
|---|---|
| Seed stocks | *Report on the source of all seed stocks or other plant material used. If applicable, state the seed stock centre and catalogue number. If plant specimens were collected from the field, describe the collection location, date and sampling procedures.* |
| Novel plant genotypes | *Describe the methods by which all novel plant genotypes were produced. This includes those generated by transgenic approaches, gene editing, chemical/radiation-based mutagenesis and hybridization. For transgenic lines, describe the transformation method, the number of independent lines analyzed and the generation upon which experiments were performed. For gene-edited lines, describe the editor used, the endogenous sequence targeted for editing, the targeting guide RNA sequence (if applicable) and how the editor was applied.* |
| Authentication | *Describe any authentication procedures for each seed stock used or novel genotype generated. Describe any experiments used to assess the effect of a mutation and, where applicable, how potential secondary effects (e.g. second site T-DNA insertions, mosiacism, off-target gene editing) were examined.* |

# ChIP-seq

## Data deposition

☒ Confirm that both raw and final processed data have been deposited in a public database such as GEO.

☐ Confirm that you have deposited or provided access to graph files (e.g. BED files) for the called peaks.

| | |
|---|---|
| Data access links<br>*May remain private before publication.* | *For "Initial submission" or "Revised version" documents, provide reviewer access links. For your "Final submission" document, provide a link to the deposited data.* |
| Files in database submission | *Provide a list of all files available in the database submission.* |
| Genome browser session<br>(e.g. UCSC) | *Provide a link to an anonymized genome browser session for "Initial submission" and "Revised version" documents only, to enable peer review. Write "no longer applicable" for "Final submission" documents.* |

## Methodology

| | |
|---|---|
| Replicates | Two biological replicates were generated for each ChIP-seq experiment |
| Sequencing depth | All sequencing depth data for each sample are detailed in Table S4 |
| Antibodies | Cell Signaling Technology (anti-H3: #4620, H3K4me3: #9751S, H3K9ac: #9649S, H3K79me2: H4K20me3: #5737S)<br>Abcam (H3K36me3: #ab9050) |
| Peak calling parameters | Peaks were called with macs2 using default parameters. |
| Data quality | All QC data including the number of peaks and FRIP scores for each sample are detailed in Table S4 |
| Software | All ChIP-seq data was processed using nf-core/chipseq v2.0.0 with default options and its associated singularity containers. All the information related to the pipeline, tools and the versions used are detailed at https://nf-co.re/chipseq/2.0.0/ |

