## [Peer Review File · Nature Ecology & Evolution]

Evolution of a distinct chromatin regulatory landscape in brown algae

Corresponding Author: Dr Susana Coelho

Version 0:

Decision Letter:

28th October 2025

Dear Susana,

Your manuscript entitled "Rewiring of chromatin regulation underlies the evolution of brown algal multicellularity" has now been seen by three reviewers, whose comments are attached. The reviewers have raised a number of concerns which will need to be addressed before we can offer publication in Nature Ecology & Evolution. We will therefore need to see your responses to the criticisms raised and to some editorial concerns, along with a revised manuscript, before we can reach a final decision regarding publication.

We therefore invite you to revise your manuscript taking into account all reviewer and editor comments. Please highlight all changes in the manuscript text file in Microsoft Word format.

* If you have not done so already please begin to revise your manuscript so that it conforms to our Article format instructions at <http://www.nature.com/natecolevol/info/final-submission>. Refer also to any guidelines provided in this letter.

* Extended Data Figures - please ensure that any supplementary figures and tables that are crucial to the manuscript's conclusions are converted into Extended Data figures and tables to increase visibility of these data. Extended Data figures and tables are online-only (present in the online PDF and full-text HTML versions of the paper), peer-reviewed display items that provide essential background to the article but are not included in the main article due to space constraints. A maximum of ten Extended Data display items (figures and tables) is permitted.

Link Redacted

Nature Ecology & Evolution is committed to improving transparency in authorship. As part of our efforts in this direction, we are now requesting that all authors identified as 'corresponding author' on published papers create and link their Open Researcher and Contributor Identifier (ORCID) with their account on the Manuscript Tracking System (MTS), prior to acceptance. ORCID helps the scientific community achieve unambiguous attribution of all scholarly contributions. You can create and link your ORCID from the home page of the MTS by clicking on 'Modify my Springer Nature account'. For more information please visit www.springernature.com/orcid.

[redacted]

Reviewer expertise:

Reviewer #1: comparative epigenomics and evolution of eukaryotes

Reviewer #2: evolution of chromatin and genome architecture

Reviewer #3: comparative genomics, sex chromosome evolution

Reviewers' comments:

Reviewer #1 (Remarks to the Author):

In this article, Vigneau and colleagues explore the evolution of chromatin in the brown algal lineage. Using comparative genomics and epigenomic analyses, they show that multicellular brown algae are divergent eukaryotes in terms of chromatin regulation. Notably, the loss of canonical heterochromatin marks appears to be ancestral to this group, while H3K79me₂, typically associated with transcription in other eukaryotes, has acquired a silencing role. The authors further demonstrate that chromatin states are conserved across the lineage: broadly conserved genes tend to share conserved histone profiles, whereas new genes are often located in repressive chromatin regions. They also characterize chromatin organization in the sex chromosomes of these species, detecting changes across the U and V chromosomes. Finally, they include a unicellular sister lineage to brown algae, which retains cytosine methylation (albeit in an unusual pattern). Overall, this is a high-quality and comprehensive dataset that provides an exciting new view of chromatin evolution in a fascinating eukaryotic clade.

Because the article is not formatted according to the Nature Ecology & Evolution style, it is currently difficult to determine which figures correspond to Extended Data. Clarifying this would help the reader, as some figures in the Supplementary Material (e.g. Fig. S5) are central and should be easily accessible, whereas others (e.g. Fig. S8) could remain in a separate supplementary file.

I have a few comments that should be addressed:

DNA methyltransferase search: use a PFAM domain search with HMMER and include a phylogenetic analysis to classify these genes, as Blastp searches are problematic for this family. There may be DNMTs other than MET1/DNMT1, DNMT3, or DRM (e.g. DNMT5, DNMT4). Please specify what is meant by "DN4MT" in Figure 1A. I assume this refers to the 4mC methyltransferase reported in *Marchantia*, but this enzyme results from lateral gene transfer from bacteria and is not representative of most eukaryotic DNMTs, including those in Heterokonta. Also, humans have three DNMT3 family members (including the catalytically inactive DNMT3L), not four. Similarly, *Arabidopsis* does not have seven MET1 orthologues. The high count may be due to inclusion of CMTs, which are a plant-specific DNMT1 subfamily, just as DRMs are a plant-specific DNMT3 subfamily. There are several previous publications classifying DNMTs in the eukaryotes, which could be helpful here (PMID 40712095, 36894681). In fact, MET1 is generally known as DNMT1 in eukaryotes.

Is the absence of H3K9 and H3K27 methylation related to substitutions in the histone tail residues in brown algae?

DOT1 expansion: there is no discussion in the main text of protein domain expansion within each family. Are these domain architectures conserved across the brown algal phylogeny? Are these derived or ancestral?

While the silencing role of H3K79me₂ is convincing, it remains unclear what regulates transposable element expression in these species. The "Repeat" category in Figure 2B may include intragenic repeats consistent with H3K36me₃ marking. It would be useful to focus on relatively young long transposable elements located in intergenic regions (to avoid overlap with protein-coding genes) and assign them to known classes (DNA, LINE, LTR, etc.) to test whether any histone modification is specifically associated with them. Another genomic feature that could be interesting to focus on are the viral insertions described for these genomes in PMID 39571576, as those should probably be in heterochromatin regions.

Line 238 – The pairwise comparisons do not seem to match phylogenetic distance: *Upin* appears closer to *Esp7* than to *Ddud* or *Dher*, yet the number of orthologues with similar patterns is lower. Perhaps I am misinterpreting the figure?

Regarding the "young" genes that are silenced and located in heterochromatic regions, could some be misannotations or fragments of transposable elements? Gene-finding pipelines often produce such artifacts. Filtering for genes that contain a known protein domain (not typical of a transposable element) or have expression in at least one sample could reduce potential noise.

Line 328 – *S. ischiensis* is not an ancestor of brown algae but their sister group, with its own evolutionary history and likely divergence from a common ancestor. More on this below.

Line 336 – Although I agree that 4mC and 6mA calls are likely false positives, the statement that they represent “less than 5% global methylation” is not very convincing. Some eukaryotes with bona fide 6mA exhibit levels below 5%, especially since 6mA is restricted to specific dinucleotide contexts on narrow genomic regions. It would be more convincing to link these negative results to the absence of 4mC DNMTs (e.g. DN4MT in Figure 1) and known eukaryotic 6mA methyltransferases (AMT1, see PMID 31722409 or <https://www.biorxiv.org/content/10.1101/2024.10.28.620566v1>), as the absence of these genes clearly argues against genuine 4mC/6mA methylation.

Line 341 – The CHG and CHH 5mC contexts are not strongly enriched in any genomic feature, yet their levels are higher in the nuclear than in the plastid genome. Since Nanopore sequencing can overestimate non-CG methylation, genome inspection might reveal whether these contexts form enriched “islands.” The expectation is that this genome shows primarily CG methylation, similar to diatoms, so confirming whether these CHG/CHH signals are artifacts (or not) would be valuable.

Line 344 – The CG methylation pattern in *S. ischiensis* resembles that of hypermethylated species, with levels around 80%, similar to vertebrates. Several eukaryotes independently evolved such hypermethylation, often anticorrelated with H3K4me3. Placing this in broader context would help readers (e.g. PMID 20395474, 31726061). The genes fall into two clusters, but are there genes completely lacking 5mCG in their bodies? Or is it the default state? Then, this pattern is not classic gene-body methylation as in plants or invertebrates, as there the transcriptional status of a gene correlates with 5mC along the gene body. Given the scarcity of this pattern across the phylogeny, it could be yet another example of convergent evolution. Because of this, it would be worth examining the DNMT repertoire of this species more thoroughly: MET1 (DNMT1) may partially explain this pattern, but as it is mainly a maintenance enzyme at CG dinucleotides, additional DNMTs could be involved for de novo activity. There is also a striking lack of correlation between H3K36me3 and 5mC in cluster 2 genes, in contrast to animals. This should be noted, as hypermethylation has arisen independently in various eukaryotic lineages and may not reflect the ancestral state.

Line 352 – I do not follow the logic of the argument that this 5mC pattern explains weak correlations between histone modifications. For example, Figure S5 suggests that H3K79me2 and H3K36me3 still show dynamic links with transcription (positive and negative correlations). Although CG methylation appears widespread, this does not preclude additional roles for histone modifications. E.g. vertebrate genomes are also globally methylated but retain strong transcriptional and histone mark variation. H3K4me3 may protect against methylation, as in other eukaryotes, rather than requiring active demethylation. If cluster 2 genes require demethylation for activation, this could be tested by examining expression across different stages (if any). Figure 5D shows that cluster 2 genes are expressed at lower levels than cluster 1, but not completely silent. Therefore, the presence of 5mC in promoters does not necessarily indicate silencing.

Line 385 – The vertebrate regulatory system includes H3K9 and H3K27me3, perhaps rephrasing this sentence and quoting other examples of hypermethylated genomes across the eukaryotes could be useful for the readers.

Line 407 – “Perceived repressive role”, here the authors could simply consider the two possibilities, that H3K79me2 is indeed repressive, or that it is a byproduct of transcription (or the lack of thereof). As with many histone modifications, steady state correlations do not always preclude causality. Good to be open about these possibilities.

Line 426 – If space allows, perhaps it would be nice to spell out the similarities with other multicellular eukaryotes a bit more in detail, especially in species with similar sexual chromosomes.

Minor points:

Title: perhaps “underlies” is a strong verb, as it is also possible that a bottleneck and simplification of hPTM in the ancestor before emergence of multicellularity was the cause of such a divergent chromatin landscape, and multicellularity evolved much later.

Line 152 – You refer to “two ectocarpales species,” but it is not clear which species are included in this taxonomic term (for the not so familiarised reader).

Line 155 – The phrase “were strongly associated with reduced gene expression” could be replaced with “were anticorrelated with gene expression” for clarity.

Figure 2A – The genome browser snapshots are difficult to read. Increasing their vertical size would help readability, as the resolution deteriorates when zoomed in.

Figure 3 – It would be helpful to remind readers what each chromatin state (S1, S2, S3) represents, as these labels are easy to forget.

Line 297 – Please explain briefly what “co-sexuality” means in this context.

Line 377 – The loss of DNA, H3K9, and H3K27 methylation is not unique among eukaryotes; similar cases occur, for example, in *Capsaspora owczarzaki* (PMID 27114036).

Reviewer #2 (Remarks to the Author):

In this study the authors compared chromatin profiles with respect to several histone modifications across several species belonging to the brown algae. This group of species is interesting because they have lost canonical mechanisms of transcriptional repression, including lost polycomb-mediated silencing and DNA methylation. After confirming the absence and presence of several chromatin modifying complexes, the authors assessed differences in histone modifications using MS and WB analyses. They further profiled the genome-wide enrichment of these modifications in males and females of five brown algal species and one outgroup. Focusing on genes, they further found that evolutionary young genes are associated with repressed or “null” chromatin signatures compared to old genes suggesting that young genes arise in heterochromatic regions. Alternatively, I think young genes may not yet have acquired the regulatory elements to strongly activate their expression, which would turn their chromatin signature into a more active state.

Furthermore, they investigated chromatin signatures associated with sex chromosomes and found that these are less

conserved than autosomes consistent with their faster rate of sequence evolution and rapid gene turnover. Comparing chromatin profiles on sex chromosome-linked genes between males and females or in a co-sexual species they found that chromatin profiles specifically on female-biased genes are altered between sexes. Finally, focusing on their chosen outgroup, they describe that, in contrast to the brown algae, *S. ischiensis* methylates its genome in the CG context consistent with the presence of MET1 gene. Interestingly, the depletion of 5mCG around promoter regions suggests a role of DNA methylation in the regulation of gene expression similar to mammalian species.

This is an interesting study that will be relevant for researchers working in the areas of chromatin and genome evolution. I have one major comment to section 2 that requires clarification:

While the enrichment of H3K4me3 and H3K9ac over TSSs and H3K36me3 over expressed gene bodies is obvious, I am confused about their discussion of the previously described repressive mark H3K79me2 and its impact on gene expression. Based on Figure S5, H3K79me2 does not show any enrichment over gene bodies (negative Log2 ChIP/H3 input) except for the outgroup. This contrasts with the statement written by the authors "H3K79me2 was broadly deposited over gene bodies..." (line 152 – 153). In addition, in contrast to the author's description (lines 155 and 202), I do not observe any clear differences in the H3K79me2 distribution over TSSs or gene bodies between the different organisms in Figure S5. Overall, it is therefore unclear to me why the authors investigate a correlation between the enrichment of this mark and transcriptional activity over annotated genes. According to their analyses in Figure 2B, this mark appears to be more enriched in intergenic regions and over repeats, which might explain its relative depletion over genes. It could be helpful to show some examples of this enrichment over these regions, different to the one's shown in Figure 2A and, if relevant, discuss the function of this mark over those genomic regions.

Other comments:

Title: To me the title does not reflect the content of the paper. It gives the impression that the paper reveals chromatin changes that drive the evolution to multicellularity, which is not tested here.

Figure 2A: The location of some of the broad peaks that were called seem to be off in the tracks of "Upin females"

There is no discussion on H4K20me3 in section 2. This mark seems to show an interesting difference between male and female *Ectocarpus* according to Figure S5

Line 202 and 203: I think it should be Figure S5?

All figure legends are a bit minimal and could be expanded to provide more information about the data shown.

Limitation of this review: I am not an expert in any of the chromatin HMM analyses that were performed.

Reviewer #3 (Remarks to the Author):

This is an interesting work studying the epigenetic landscape (histone modification, HPTM and DNA methylation) patterns of six brown algae species, a lineage that evolved independently multicellularity. Thanks to the previous work of ENCODE and modENCODE, we have formed a general perspective of very conserved function and genomic distribution of histone modifications from yeast to human. But the brown algae seems to hold a different story: in this work, the authors found that all studied species but *S. ischiensis* or the Phaeophyceae lineage has lost the critical enzyme responsible for DNA methylation and Polycomb proteins responsible for the repressive histone mark H3K27me3. The canonical active histone mark H3K79me is instead associated with the gene silencing. By contrast, the active histone modifications are conserved in their function and genomic distribution. Finally the author also compared the HPTMs patterns across species, and between sexes when possible, and revealed complex changes associated with transitions between co-sexuality vs. dioecy, as well as the patterns between HPTMs vs. sex-biased genes. These results indicate that unexpectedly, brown algae species may adopt a different epigenetic strategy to regulate the gene expression, compared to human and yeasts. I think the first part of the work, i.e., change of epigenetic landscape, including the chromatin states and responsible proteins, are novel and clear. But the latter conclusions of the work, that is, the association of these epigenetic patterns with sexual reproduction and sex chromosomes, are a bit complicated to gain clear message by the reader. I detail my comments below:

1. A key assumption of this work is that expansion of DOT1, the gene responsible for the H3K79me modification in brown algae, maybe associated with the reversed association of this HPTM with active genes. This is directly related to one of the key findings of this work, so I think the DOT1 gene family pattern merits a panel in the main figure. In addition, how are these distinct families expressed in brown algae? It has been discussed in the final part of the work that expansion and diversification of this gene family is critical for the brown algae specific epigenetic regulation. But how exactly? One would not expect that all gene copies of DOT1 would have the similar function of regulating H3K79me. So some clues might be gained from comparing these gene copies' coding sequences (or their critical protein domains), as well as their different expression patterns, if any.

2. Several important results of the work lack statistical tests or turned out to be vague. These include:

- 1) Line 153, the association of H3K79me2 binding levels with gene expression levels. Note there are also no figure or table linked to this very important conclusion. And how did the authors define 'highly expressed genes' at the same line?
- 2) Line 157, ..overall correlations were weaker..., here need to show the exact values and matrices

3) Line 278, Figure 4A, ...showing reduced proportions..and increased proportions..are these difference significant between sex chromosomes and autosomes?

3. Figure 2C, here S14 and S12 chromatin states were further defined as 'active' states, and they were highlighted as in the species *Upin*, that they show low gene expression levels and also bound by H3K79me2. This is confusing if these 'active' chromatin states in fact have a silencing function on gene expression? In addition, how is the H4K20me3 binding pattern in these states? Also this figure, the expression levels of *Upin* at state S14 and S12 are close to zero in white color. But this is different from Figure S3C, where the same two values look like higher than 0. Could you please clarify these points?

4. Line 283-296, this part of the result, i.e., the comparison of chromatin states between sexes, is detached from previous results. Because emergence of dioecy or turnover of sex chromosomes occurred after loss of DNA methylation or gain of H3K79me function. Given their recent origin, there are many species specific pattern of chromatin states between sexes. For example, some species involve changes only in female-biased genes, some involve both male- and female-biased genes. However these descriptive patterns seem to lack explanations or hypotheses to be accounted for. I suggest author could refer to the works in plants (Zemp et al. 2016 Nat. Plants) and also their previous work of comparing expression changes between co-sexuality vs. dioecious algae (Cossard et al. 2022 Nat. Ecol. Evol) to build up more links between the results and explanations related to phenotypes/ecological traits of different species.

*****END*****

Version 1:

Decision Letter:

20th January 2026

Dear Susana,

Thank you for submitting your revised manuscript "Evolution of a distinct chromatin regulatory landscape in brown algae" (NATECOLEVOL-25093379A). It has now been seen again by the original reviewers and their comments are below. The reviewers find that the paper has improved in revision, and therefore we'll be happy in principle to publish it in *Nature Ecology & Evolution*, pending minor revisions to satisfy the reviewers' final requests and to comply with our editorial and formatting guidelines.

Thank you again for your interest in *Nature Ecology & Evolution*. Please do not hesitate to contact me if you have any questions.

[redacted]

Reviewer #1 (Remarks to the Author):

The authors have convincingly addressed my previous comments, and I believe that the revisions have strengthened the manuscript overall. The improved formatting of the figures and extended data figures is effective in maximising clarity, and I also appreciated the expanded discussion of U/V sex chromosomes in other systems.

I agree with the authors that, in its current form, the association between hPTMs and TE classes is not especially informative. A comprehensive, species-specific annotation of TEs would be required to address this properly, which would clearly represent a substantial effort and is more appropriate for a potential future dedicated study. While my initial hope was that the curated *Ectocarpus* TE library might allow identification of sufficiently abundant or young TEs in the other species, it is clear that this group is highly diverse, having diverged approximately 450 million years ago. In this context, I agree with the authors that these preliminary analyses are better omitted from the main manuscript. The same reasoning applies to the viral insert analyses.

I have only one minor remaining comment:

Line 619: While H3K36me2 has been shown to recruit DNMT3A in mammals (PMID: 31485078), this mark is predominantly intergenic rather than a gene-body modification. The authors may instead wish to refer to H3K36me3, which has been shown to correlate with gene-body 5mCG in both vertebrates and invertebrates.

Alex de Mendoza

Reviewer #2 (Remarks to the Author):

The motivation to look into a role of H3K79me2 over gene bodies seems clearer to me now. To illustrate to what extent this mark contributes to gene repression across all genes in the genome, I think it is important that the authors mention in the text that they perform the analyses over a subset of genes (those that overlap H3K79me2 peaks as stated in the response letter). They should also add for each species the number of genes examined the same way as for *S. ischiensis* (753/21187; 3.5%). Along these lines and following up on their discussion on a potential contribution of this mark in repressing transposable elements, is it possible that genes are not specifically targeted by H3K79me2, but rather that the mark accumulates over genes due to their proximity to repressed transposable elements? If the authors agree with this hypothesis, a short discussion could be added to the manuscript.

Other comments:

Line 194: I guess it should be "transcription start site" instead of "translational start site"

Reviewer #3 (Remarks to the Author):

The authors have addressed all my previous questions and comments with satisfaction. I don't have further questions

Reviewers' comments:

Reviewer #1 (Remarks to the Author):

In this article, Vigneau and colleagues explore the evolution of chromatin in the brown algal lineage. Using comparative genomics and epigenomic analyses, they show that multicellular brown algae are divergent eukaryotes in terms of chromatin regulation. Notably, the loss of canonical heterochromatin marks appears to be ancestral to this group, while H3K79me₂, typically associated with transcription in other eukaryotes, has acquired a silencing role. The authors further demonstrate that chromatin states are conserved across the lineage: broadly conserved genes tend to share conserved histone profiles, whereas new genes are often located in repressive chromatin regions. They also characterize chromatin organization in the sex chromosomes of these species, detecting changes across the U and V chromosomes. Finally, they include a unicellular sister lineage to brown algae, which retains cytosine methylation (albeit in an unusual pattern). Overall, this is a high-quality and comprehensive dataset that provides an exciting new view of chromatin evolution in a fascinating eukaryotic clade. Because the article is not formatted according to the Nature Ecology & Evolution style, it is currently difficult to determine which figures correspond to Extended Data. Clarifying this would help the reader, as some figures in the Supplementary Material (e.g. Fig. S5) are central and should be easily accessible, whereas others (e.g. Fig. S8) could remain in a separate supplementary file.

Re: We are grateful to the reviewer for the kind assessment of this work. Regarding the figure names, we have renamed the current supplementary figures accordingly:

Extended Data Figures

Extended Data Figure 1 (previously Fig S1 gene trees)

Extended Data Figure 2 (previously Fig S3 male hiHMM)

Extended Data Figure 3 (previously Fig S5 metaplots)

Extended Data Figure 4 (previously Fig S7 emission states)

Extended Data Figure 5 (previously Fig S9 signature TPM)

Extended Data Figure 6 (previously Fig S11 signature gene age)

Extended Data Figure 7 (previously Fig S12 signature orphan)

Extended Data Figure 8 (previously Fig S13 signature uniformity)

Extended Data Figure 9 (previously Fig S14 Sisc DNAm)

Supplementary Figure

Supplementary Figure 1 (previously Fig S2 western blots)

Supplementary Figure 2 (previously Fig S4 chip replicates)

Supplementary Figure 3 (previously Fig S6 transition matrix)

Supplementary Figure 4 (previously Fig S8 RNA-seq correlation)

Supplementary Figure 5 (previously Fig S10 1:1 analysis)

I have a few comments that should be addressed:

DNA methyltransferase search: use a PFAM domain search with HMMER and include a phylogenetic analysis to classify these genes, as Blastp searches are problematic for this family. There may be DNMTs other than MET1/DNMT1, DNMT3, or DRM (e.g. DNMT5, DNMT4).

Re: We thank the reviewer for this important suggestion. In the revised analysis, we complemented our initial OrthoFinder-based homology inference with a domain-centric strategy using Pfam HMMs, as recommended. Specifically, we searched all brown algal proteomes and the outgroup species for the C-5 cytosine-specific DNA methyltransferase catalytic domain (PF00145) and the associated N6_N4_Mtase domain (PF01555), which is characteristic of C4-type DNA methyltransferases. All sequences containing these domains were then aligned with representative DNMT1/MET1, DNMT3/DRM, DNMT2/TRDMT, and DNMT5 proteins from human, *Arabidopsis*, and/or *Cryptococcus neoformans*.

These additional phylogenetic analyses corroborate our original OrthoFinder results. Although the outgroup species contain DNMT1-like sequences, no *bona fide* DNMT1, DNMT3/DRM, DMT4 nor DNMT5 orthologs are detected in brown algae. The DNMT hits in brown algae consistently cluster within the DNMT2/TRDMT clade, indicating that tRNA methylation activity is likely retained across the Phaeophyceae. These new phylogenetic results are included as new **Extended Data Figure 1**, and **Extended Supplementary Data** with the results and methods sections revised accordingly.

Please specify what is meant by “DN4MT” in Figure 1A. I assume this refers to the 4mC methyltransferase reported in *Marchantia*, but this enzyme results from lateral gene transfer from bacteria and is not representative of most eukaryotic DNMTs, including those in Heterokonta. Also, humans have three DNMT3 family members (including the catalytically inactive DNMT3L), not four. Similarly, *Arabidopsis* does not have seven MET1 orthologues. The high count may be due to inclusion of CMTs, which are a plant-specific DNMT1 subfamily, just as DRMs are a plant-specific DNMT3 subfamily. There are several previous publications classifying DNMTs in the eukaryotes, which could be helpful here (PMID 40712095, 36894681). In fact, MET1 is generally known as DNMT1 in eukaryotes.

Re: We appreciate the reviewer’s careful observation of our data. We have responded and improved the revised manuscript as follows: (1) “DN4MT” in **Figure 1A** does indeed refer to the 4mC methyltransferases in *Marchantia* and was included to ensure a comprehensive analysis as possible. By performing a similar phylogenetic analysis as above, we corroborated our OrthoFinder-based homology inference and confirmed that 4mC methyltransferases are undetectable in brown algae. (2) We have revised our family nomenclature in **Figure 1A** to reflect the usage suggested as follows: MET1 \equiv DNMT1, DRM proteins = DNMT3-like (plant lineage-specific), CMT proteins are now annotated as a new separate subfamily rather forming part of the DNMT1/MET1 group. (3) We have corrected copy-number annotations for animals and plants in **Fig. 1A** as follows: (a) The four members of the Human DNMT family included an additional uncharacterised gene with a DNMT3-like domain (Uniprot ID = A0A096LPK6). We have removed this gene from the count and only report the three *bona fide* DNMT genes. (b) *Arabidopsis* DNMT family sizes have been split into DNMT1/MET1, CMT, and DRM subgroups. (4) We now also cite the DNMT papers suggested (PMID 40712095; PMID 36894681).

Is the absence of H3K9 and H3K27 methylation related to substitutions in the histone tail residues in brown algae?

Re: We have now added a panel **Supplemental Figure 1A** (previously Fig. S2) showing an alignment of the N-terminal tails of several brown algal histone H3 sequences alongside histone H3.1, H3.3 and the atypical H3.10 from *Arabidopsis*. These results show that the N-terminal tails are highly conserved and that the lack of H3K9 and H3K27 methylation is not caused by changes at or around these residues.

DOT1 expansion: there is no discussion in the main text of protein domain expansion within each family. Are these domain architectures conserved across the brown algal phylogeny? Are these derived or ancestral?

Re: In response, we have now performed an in-depth phylogenetic analysis of each DOT1 subfamily (**Extended Data Figure 1B**). This analysis confirms that one subfamily (which we designate as DOT1.1) represents the ancestral copy given its homology with yeast DOT1 and human DOT1L, whereas the other four represent derived subfamilies. The domain scans and whole gene alignment-derived phylogenies we generated highlight novel domain associations and variable domain architecture of the DOT1 subfamilies across brown algae. We further show differential expression patterns of DOT1 genes using previously published *Ectocarpus* transcriptomes (**Extended Data Figure 1D**), which further supports the functional diversification of the DOT1 gene family.

While the silencing role of H3K79me2 is convincing, it remains unclear what regulates transposable element expression in these species. The “Repeat” category in Figure 2B may include intragenic repeats consistent with H3K36me3 marking. It would be useful to focus on relatively young long transposable elements located in intergenic regions (to avoid overlap with protein-coding genes) and assign them to known classes (DNA, LINE, LTR, etc.) to test whether any histone modification is specifically associated with them.

Re: We thank the reviewer for this suggestion. We agree that identifying hPTMs associated with specific TE classes would be informative. However, while a high-quality curated TE library exists for *Ectocarpus* (as reported in our recent work [Dinatale et al., (2025) *Genome Biology*]), properly curated TE annotations for the other species analysed here are not available. The *de novo* annotations we have generated result in repeat landscapes that are overwhelmingly dominated by unclassified or “unknown” elements, making class-level analyses unreliable. Developing curated, lineage-specific TE libraries comparable to that of *Ectocarpus* would require extensive manual curation and is beyond the scope of the present study.

Our previous work in *Ectocarpus* showed that H3K79me2 and sRNAs have a small but significant preference for intact TEs, although we did not observe strong enrichment for most classifiable TE families (Dinatale et al., 2025, *Genome Biology*). In our current study, we examined intact young TEs across species (defined as long TEs with <20% divergence from their consensus sequence) and again found no strong preference of H3K79me2 or any other mark for specific TE classes (see **Peer Review Figure 1**), further corroborating results shown in Fig. 2B. These genomes are characterised by a highly interspersed organisation of TEs, where intergenic space is limited, gene density is high, and TE families are broadly and evenly distributed (see new tracks in **Figure 2A**). Consequently, most chromatin states appear enriched at both repeated and intergenic regions largely because these genomic features frequently co-occur. While this analysis does not fully resolve how TEs are definitively silenced across all these species, our data from *Ectocarpus* suggest that sRNAs together with H3K79me2 may contribute to TE repression (Dinatale et al., 2025, *Genome Biology*), which would be interesting to tackle in a future cross-comparative study.

Peer Review Figure 1. Proportion of different classes of TEs marked with the different hPTMs across the six species analysed in this study.

Another genomic feature that could be interesting to focus on are the viral insertions described for these genomes in PMID 39571576, as those should probably be in heterochromatin regions.

Re: This is an interesting point and we thank the reviewer for this idea. Although we believe that a detailed analysis of the inserted viral element is out of the scope of the manuscript (it is actually the object of investigation in the context of another project in our lab), we did investigate the viral insert (EVEb) in the *Ectocarpus* species investigated here and found that the viral insert is marked by repressive chromatin signatures (see **Peer Review Figure 2**, below), suggesting it is silenced and in a heterochromatic state. We would prefer not to include this analysis but can of course add it if the editor and reviewer think this is an important point. We believe that the TE analysis above is sufficient to establish the relationship between repressive chromatin states and non-host genetic elements.

Peer Review Figure 2. Chromatin signatures in the viral insert (EVEb) compared to other regions of the genome in *Ectocarpus* (male). The number of genes used in this comparison is indicated under (“n”).

Line 238 – The pairwise comparisons do not seem to match phylogenetic distance: Upin appears closer to Esp7 than to Ddud or Dher, yet the number of orthologues with similar patterns is lower. Perhaps I am misinterpreting the figure?

Re: This observation (of partial correspondence with phylogenetic distance) was made after considering the observed number of orthologues with the same signature, against randomly shuffled data. This is because the raw number of orthologues with the same signature may be confounded by genome-wide differences in the composition of chromatin signatures (which is accounted for by the permutation test). Yet, since the correlation with phylogenetic distance is indeed not very strong, we modified the phrasing to “correlated, in part, with phylogenetic distance”.

Regarding the “young” genes that are silenced and located in heterochromatic regions, could some be misannotations or fragments of transposable elements? Gene-finding pipelines often produce such artifacts. Filtering for genes that contain a known protein domain (not typical of a transposable element) or have expression in at least one sample could reduce potential noise.

Re: We thank the reviewer for this suggestion. We re-analysed the data after removing genes with weak transcriptomic evidence (mean TPM < 1) and have updated **Figs 3C, 3D, Extended Data Figure 6 and Extended Data Figure 7**. The same pattern holds. We chose not to remove transcriptionally active genes lacking domain annotations, as many genes unique to brown algae have no detectable homology to other lineages, which is to be expected given their distinct evolutionary history.

Line 328 – *S. ischiensis* is not an ancestor of brown algae but their sister group, with its own evolutionary history and likely divergence from a common ancestor. More on this below.

Re: The title to this section has been changed accordingly.

Line 336 – Although I agree that 4mC and 6mA calls are likely false positives, the statement that they represent “less than 5% global methylation” is not very convincing. Some eukaryotes with bona fide 6mA exhibit levels below 5%, especially since 6mA is restricted to specific dinucleotide contexts on narrow genomic regions. It would more convincing to link these negative results to the absence of 4mC DNMTs (e.g. DN4MT in Figure 1) and known eukaryotic 6mA methyltransferases (AMT1, see PMID 31722409 or <https://www.biorxiv.org/content/10.1101/2024.10.28.620566v1>), as the absence of these genes clearly argues against genuine 4mC/6mA methylation.

Re: We thank the reviewer for this insightful suggestion. We have now extended our analysis and also looked for eukaryotic 6mA methyltransferases (AMT1 and AMT6/7) as described in the recommended publication and added this to **Figure 1A**. We indeed find no evidence for 6mA methyltransferases across

both the brown algae and their sister ochrophytina lineage. This more convincingly links the residual “background” levels of 6mA to the absence of the known enzymes, as is the case for 4mC. The text has been updated accordingly.

Line 341 – The CHG and CHH 5mC contexts are not strongly enriched in any genomic feature, yet their levels are higher in the nuclear than in the plastid genome. Since Nanopore sequencing can overestimate non-CG methylation, genome inspection might reveal whether these contexts form enriched “islands.” The expectation is that this genome shows primarily CG methylation, similar to diatoms, so confirming whether these CHG/CHH signals are artifacts (or not) would be valuable.

Re: We had indeed looked at this and observed no obvious patterns or enriched patterns in the genome for these forms of methylation. For transparency, we have now included the 5mCHH and 5mCHG tracks in **Figure 5A** and raise the point that while the signals may be genuine, they do not appear to have a specific function nor enrichment in *Schizocladia*.

Line 344 – The CG methylation pattern in *S. ischiensis* resembles that of hypermethylated species, with levels around 80%, similar to vertebrates. Several eukaryotes independently evolved such hypermethylation, often anticorrelated with H3K4me3. Placing this in broader context would help readers (e.g. PMID 20395474, 31726061).

Re: This is a nice suggestion and we now cite this work and use it to expand our discussion.

The genes fall into two clusters, but are there genes completely lacking 5mCG in their bodies? Or it is the default state? Then, this pattern is not classic gene-body methylation as in plants or invertebrates, as there the transcriptional status of a gene correlates with 5mC along the gene body. Given the scarcity of this pattern across the phylogeny, it could be yet another example of convergent evolution. Because of this, it would be worth examining the DNMT repertoire of this species more thoroughly: MET1 (DNMT1) may partially explain this pattern, but as it is mainly a maintenance enzyme at CG dinucleotides, additional DNMTs could be involved for de novo activity. There is also a striking lack of correlation between H3K36me3 and 5mC in cluster 2 genes, in contrast to animals. This should be noted, as hypermethylation has arisen independently in various eukaryotic lineages and may not reflect the ancestral state.

Re: We thank the reviewer for raising these important points. We had not explicitly highlighted this before, but the reviewer is correct to note that, in addition to the strong differences in promoter methylation, 5mCG levels increase modestly across the gene body of cluster 1 genes. In contrast, cluster 2 genes show no enrichment of 5mCG within gene bodies relative to flanking regions. To clarify this further, we now include a plot showing 5mCG profiles across genes grouped by expression level (new panel **Fig. 5E**). This analysis confirms that although expressed genes show a slight elevation of gene-body 5mCG, the increase is uniform across expression quantiles, indicating that the pattern does not scale with transcript abundance. This contrasts with classic gene-body methylation in plants and animals, where 5mC generally correlates with expression levels (e.g., Zilberman 2017, Nat. Rev. Genet.; Bewick & Schmitz 2017, Genome Biol.).

Our revised phylogenetic analysis (**Fig. 1A**; **Extended Data Figure 1A**) confirms that *S. ischiensis* encodes only a single DNMT1 enzyme and lacks DNMT3/DRM- or DNMT5-like de novo methyltransferases. This strongly suggests that DNMT1 alone maintains the observed CG methylation landscape and that de novo 5mCG establishment occurs at very low levels involving mechanisms distinct from canonical DNMT3/DRM pathways, in part explaining the poor correlation with elevation of H3K36me3.

Together, these additions clarify that gene-body 5mCG may be present in *S. ischiensis* but is only weakly associated with transcription, and may be loosely linked to H3K36me3, reflecting a lineage-specific methylation system rather than an ancestral eukaryotic state.

Line 352 – I do not follow the logic of the argument that this 5mC pattern explains weak correlations between histone modifications. For example, Figure S5 suggests that H3K79me2 and H3K36me3 still show dynamic links with transcription (positive and negative correlations). Although CG methylation appears widespread, this does not preclude additional roles for histone modifications. E.g. vertebrate genomes are also globally methylated but retain strong transcriptional and histone mark variation. H3K4me3 may protect against

methylation, as in other eukaryotes, rather than requiring active demethylation. If cluster 2 genes require demethylation for activation, this could be tested by examining expression across different stages (if any). Figure 5D shows that cluster 2 genes are expressed at lower levels than cluster 1, but not completely silent. Therefore, the presence of 5mC in promoters does not necessarily indicate silencing.

Re: We thank the reviewer for the thoughtful critique. Our original statement referred specifically to the species-wide HiHMM model and the chromatin signatures derived from it. To avoid confusion, we have now revised the phrasing to focus solely on the phenomenon of promoter demethylation and its potential ancestral role early in brown algal evolution. We also now acknowledge in the Discussion that promoter hypomethylation may arise through H3K4me3-mediated protection from DNA methylation and cite relevant work. Thus, the presence of H3K4me3 in cluster 1 promoters may indeed reflect passive exclusion of methylation rather than active demethylation. We agree that it would be informative to examine gene expression across developmental stages. However, *Schizocladia ischiensis* is a very simple filamentous alga that lacks clear morphological differentiation, making such analyses implausible in this system.

Line 385 – The vertebrate regulatory system includes H3K9 and H3K27me3, perhaps rephrasing this sentence and quoting other examples of hypermethylated genomes across the eukaryotes could be useful for the readers.

Re: As mentioned above, the relevant papers have now been cited and the sentence rephrased.

Line 407 – “Perceived repressive role”, here the authors could simply consider the two possibilities, that H3K79me2 is indeed repressive, or that it is a byproduct of transcription (or the lack of thereof). As with many histone modifications, steady state correlations do not always preclude causality. Good to be open about these possibilities.

Re: As nicely suggested, we now raise the plausibility of both possibilities in the revised version.

Line 426 – If space allows, perhaps it would be nice to spell out the similarities with other multicellular eukaryotes a bit more in detail, especially in species with similar sexual chromosomes.

Re: We thank the reviewer for this helpful suggestion. We have expanded the Discussion to more clearly articulate parallels with other multicellular eukaryotes. In particular, we now note that the *Marchantia* UV sex chromosomes also exhibit a strongly heterochromatic landscape, suggesting that repressive chromatin may be a shared hallmark of UV sex chromosomes across distantly related lineages.

Minor points:

Title: perhaps “underlies” is a strong verb, as it is also possible that a bottleneck and simplification of hPTM in the ancestor before emergence of multicellularity was the cause of such a divergent chromatin landscape, and multicellularity evolved much later.

Re: We agree that the title may not encompass all scenarios, so we have changed to a more neutral title: “*Brown algal emergence involved extensive rewiring of chromatin regulation*”

Line 152 – You refer to “two ectocarpales species,” but it is not clear which species are included in this taxonomic term (for the not so familiarised reader).

Re: This has now been rephrased to explicitly mention the two species.

Line 155 – The phrase “were strongly associated with reduced gene expression” could be replaced with “were anticorrelated with gene expression” for clarity.

Re: Fixed.

Figure 2A – The genome browser snapshots are difficult to read. Increasing their vertical size would help readability, as the resolution deteriorates when zoomed in.

Re: To improve readability, we have increased the vertical size of the genome browser tracks in **Fig. 2A** as well as **Extended Data Figure 2A**. These panels now display the signal profiles more clearly, and the resolution is preserved in the final exported figures. We hope this adjustment adequately addresses the reviewer's concern.

Figure 3 – It would be helpful to remind readers what each chromatin state (S1, S2, S3) represents, as these labels are easy to forget.

Re: As requested, a key has now been added to Fig 2 to help remind the reader what each state represents.

Line 297 – Please explain briefly what “co-sexuality” means in this context.

Re: We have now clarified this part by stating more explicitly that we focus on “*chromatin reconfigurations associated with the transition from separate sexes to co-sexuality*”.

Line 377 – The loss of DNA, H3K9, and H3K27 methylation is not unique among eukaryotes; similar cases occur, for example, in *Capsaspora owczarzaki* (PMID 27114036).

Re: we are now more explicit by mentioning that the concurrent loss of these pathways is unprecedented “*among complex multicellular eukaryotes*”.

Reviewer #2 (Remarks to the Author):

In this study the authors compared chromatin profiles with respect to several histone modifications across several species belonging to the brown algae. This group of species is interesting because they have lost canonical mechanisms of transcriptional repression, including lost polycomb-mediated silencing and DNA methylation. After confirming the absence and presence of several chromatin modifying complexes, the authors assessed differences in histone modifications using MS and WB analyses. They further profiled the genome-wide enrichment of these modifications in males and females of five brown algal species and one outgroup. Focusing on genes, they further found that evolutionary young genes are associated with repressed or “null” chromatin signatures compared to old genes suggesting that young genes arise in heterochromatic regions. Alternatively, I think young genes may not yet have acquired the regulatory elements to strongly activate their expression, which would turn their chromatin signature into a more active state.

Furthermore, they investigated chromatin signatures associated with sex chromosomes and found that these are less conserved than autosomes consistent with their faster rate of sequence evolution and rapid gene turnover. Comparing chromatin profiles on sex chromosome-linked genes between males and females or in a co-sexual species they found that chromatin profiles specifically on female-biased genes are altered between sexes. Finally, focusing on their chosen outgroup, they describe that, in contrast to the brown algae, *S. ischiensis* methylates its genome in the CG context consistent with the presence of MET1 gene. Interestingly, the depletion of 5mCG around promoter regions suggests a role of DNA methylation in the regulation of gene expression similar to mammalian species.

This is an interesting study that will be relevant for researchers working in the areas of chromatin and genome evolution. I have one major comment to section 2 that requires clarification:

While the enrichment of H3K4me3 and H3K9ac over TSSs and H3K36me3 over expressed gene bodies is obvious, I am confused about their discussion of the previously described repressive mark H3K79me2 and its impact on gene expression. Based on Figure S5, H3K79me2 does not show any enrichment over gene bodies (negative Log2 ChIP/H3 input) except for the outgroup. This contrasts with the statement written by the authors “H3K79me2 was broadly deposited over gene bodies...” (line 152 – 153). In addition, in contrast

to the author's description (lines 155 and 202), I do not observe any clear differences in the H3K79me2 distribution over TSSs or gene bodies between the different organisms in Figure S5. Overall, it is therefore unclear to me why the authors investigate a correlation between the enrichment of this mark and transcriptional activity over annotated genes. According to their analyses in Figure 2B, this mark appears to be more enriched in intergenic regions and over repeats, which might explain its relative depletion over genes. It could be helpful to show some examples of this enrichment over these regions, different to the one's shown in Figure 2A and, if relevant, discuss the function of this mark over those genomic regions.

Re: We thank the reviewer for raising this discrepancy. H3K79me2 marks only a subset of genes in each genome, ranging from approximately 3–50% depending on the species. Our initial metaplots were generated across all genes, so the large proportion of unmarked genes that contributed a null signal artificially reduced the enrichment profile, leading to apparent negative values.

To address this, we now provide revised metaplots that include only genes overlapping an H3K79me2 peak, further grouped by gene expression quantiles (new panel **Fig. 2E**). These plots clarify that: (1) when present, H3K79me2 is enriched along gene bodies; (2) the anticorrelation between gene-body H3K79me2 and transcript abundance is largely restricted to the Ectocarpales; and (3) across all species, the strongest repressive association of H3K79me2 occurs at the TSS, where expressed genes consistently show clear depletion of the mark relative to the gene body.

To strengthen these conclusions, we additionally (1) include standard-error shading around each metaplot profile to illustrate robust, non-overlapping signals across expression quantiles, and (2) perform overlap enrichment tests comparing genes with H3K79me2-marked TSSs to gene expression quantiles (new panel **Fig. 1G**). These analyses statistically support that H3K79me2-marked TSSs are enriched among the lowest expression quantiles (1-2) or genes with no expression (0), while highly expressed genes are significantly enriched among those lacking the mark.

As requested, we also provide new genome browser tracks of ChIP-seq data across large (600kb) chromosomal regions to give a better impression of how the hPTMs manifest across the different species (revised **Figure 2A** and **Extended Data 2A**). These tracks better convey how H3K79me2 coats gene bodies but also extends into intergenic regions given can form broad domains (this is most evident in *Ectocarpus* for example). The brown algal genomes we investigated are characterised by a highly interspersed organisation of TEs, where intergenic space is limited, gene density is high (bar *Undaria*), and TE families are broadly and evenly distributed (see new **Figure 2A**). Consequently, most chromatin states appear enriched at both repeats and intergenic regions largely because these genomic features frequently co-occur rather than because they function specifically in these regions repressing repeats (see response to Reviewer 1 and Peer Review Figure 1 for more details on TE repression).

Other comments:

Title: To me the title does not reflect the content of the paper. It gives the impression that the paper reveals chromatin changes that drive the evolution to multicellularity, which is not tested here.

Re: We agree that the previous title was misleading and have changed to a more neutral title that better reflects our findings: "*Brown algal emergence involved extensive rewiring of chromatin regulation*"

Figure 2A: The location of some of the broad peaks that were called seem to be off in the tracks of "Upin females"

Re: In response to a comment from Reviewer 1, we have replaced the genome browser tracks showing much broader regions (see **Figure 2A**).

There is no discussion on H4K20me3 in section 2. This mark seems to show an interesting difference between male and female *Ectocarpus* according to Figure S5

Re: We thank the reviewer for pointing this out. We have now addressed this by adding a brief discussion of H4K20me3 to Section 2. Although H4K20me3 shows a modest difference between male and female *Ectocarpus* in Figure S5, the signal is very weak and close to background levels relative to the other profiled hPTMs. We now note this observation in the text but deliberately avoid overinterpretation, as the enrichment is not sufficiently robust to support a strong biological conclusion.

Line 202 and 203: I think it should be Figure S5?

Re: Yes, this has been fixed.

All figure legends are a bit minimal and could be expanded to provide more information about the data shown.

Re: As requested, we have expanded the figure legends to include more information.

Limitation of this review: I am not an expert in any of the chromatin HMM analyses that were performed.

Reviewer #3 (Remarks to the Author):

This is an interesting work studying the epigenetic landscape (histone modification, HPTM and DNA methylation) patterns of six brown algae species, a lineage that evolved independently multicellularity. Thanks to the previous work of ENCODE and modENCODE, we have formed a general perspective of very conserved function and genomic distribution of histone modifications from yeast to human. But the brown algae seems to hold a different story: in this work, the authors found that all studied species but *S. ischiensis* or the Phaeophyceae lineage has lost the critical enzyme responsible for DNA methylation and Polycomb proteins responsible for the repressive histone mark H3K27me3. The canonical active histone mark H3K79me is instead associated with the gene silencing. By contrast, the active histone modifications are conserved in their function and genomic distribution. Finally the author also compared the HPTMs patterns across species, and between sexes when possible, and revealed complex changes associated with transitions between co-sexuality vs. dioecy, as well as the patterns between HPTMs vs. sex-biased genes. These results indicate that unexpectedly, brown algae species may adopt a different epigenetic strategy to regulate the gene expression, compared to human and yeasts. I think the first part of the work, i.e., change of epigenetic landscape, including the chromatin states and responsible proteins, are novel and clear. But the latter conclusions of the work, that is, the association of these epigenetic patterns with sexual reproduction and sex chromosomes, are a bit complicated to gain clear message by the reader. I detail my comments below:

1. A key assumption of this work is that expansion of DOT1, the gene responsible for the H3K79me modification in brown algae, maybe associated with the reversed association of this HPTM with active genes. This is directly related to one of the key findings of this work, so I think the DOT1 gene family pattern merits a panel in the main figure. In addition, how are these distinct families expressed in brown algae?

Re: We thank the reviewer for this recommendation. We now plot the expression patterns of DOT1 subfamilies using published transcriptomic data from various developmental stages of both male and female *Ectocarpus* (**Extended Data Figure 1**). We demonstrate that (1) members within the same DOT1 subfamily do not cluster in expression, suggesting distinct roles in development, and (2) a majority of DOT1 genes are upregulated in the multicellular stages compared to unicellular stages. We would prefer to keep it as an extended data figure, as the main figure is already quite packed – however we can easily transfer it to the main figure if you reviewer thinks this is an important point.

It has been discussed in the final part of the work that expansion and diversification of this gene family is critical for the brown algae specific epigenetic regulation. But how exactly? One would not expect that all gene copies of DOT1 would have the similar function of regulating H3K79me. So some clues might be

gained from comparing these gene copies' coding sequences(or their critical protein domains), as well as their different expression patterns, if any.

Re: Indeed, we also suspected that the expansion and diversification of this gene family may be linked to the presence of new protein domains and structural features. Our new in-depth analysis of the domain architecture of DOT1 genes across brown algae (**Extended Data Figure 1**) has revealed new domain associations and variable domain architecture of the DOT1 subfamilies across brown algae.

2. Several important results of the work lack statistical tests or turned out to be vague. These include:
1) Line 153, the association of H3K79me2 binding levels with gene expression levels. Note there are also no figure or table linked to this very important conclusion. And how did the authors define 'highly expressed genes' at the same line?

Re: We thank the review for raising these important points and we are happy to provide new data to clarify them. To strengthen our conclusions, we have (1) included standard-error shading around the metaplot profile of H3K79me2 to illustrate robust, non-overlapping signals across expression quantiles (new panel **Fig. 2E**), and (2) perform enrichment tests comparing genes with H3K79me2-marked TSSs to gene expression quantiles (new panel **Fig. 2G**). These analyses statistically support that H3K79me2-marked TSSs are enriched among the lowest expression quantiles (1-2) or genes with no expression (0), while highly expressed genes are significantly enriched among those lacking the mark. We have refrained from using the term "highly expressed" and replaced this with more precise reference to expression quantiles, which were defined by grouping genes based into 4 groups from high to low expression, with an additional category (0) containing genes with no expression at all (i.e. TPM = 0).

2) Line 157, ..overall correlations were weaker.., here need to show the exact values and matrices

Re: Our intention was to convey that the hPTM signals in *S. ischiensis* were weaker than in the other species. We have now rephrased this section to state this more clearly.

3) Line 278, Figure 4A, ...showing reduced proportions..and increased proportions..are these difference significant between sex chromosomes and autosomes?

Re: We now report statistics for the overall differences in proportions after applying a Chi-squared test. The test statistics has been added to the figure and the figure legend.

3. Figure 2C, here S14 and S12 chromatin states were further defined as 'active' states, and they were highlighted as in the species *U. pinna*, that they show low gene expression levels and also bound by H3K79me2. This is confusing if these 'active' chromatin states in fact have a silencing function on gene expression? In addition, how is the H4K20me3 binding pattern in these states? Also this figure, the expression levels of *U. pinna* at state S14 and S12 are close to zero in white color. But this is different from Figure S3C, where the same two values look like higher than 0. Could you please clarify these points?

Re: We thank the reviewer for raising this point, which indeed required clearer explanation. In Figure 2C, chromatin signatures are ordered by their mean expression across all samples (i.e., sexes and species combined). This ordering is used solely for readability and does not imply that a given signature acts uniformly in every species or context. Consequently, signatures that include activation-associated marks (e.g., H3K4me3 or H3K9ac) may nevertheless exhibit low expression in a particular species if additional repressive features, such as H3K79me2, are also present. It is important to emphasise that chromatin signatures reflect combinations of several underlying HiHMM states, and genes may receive more than one state assignment across their locus. Because the model captures relative enrichment patterns that differ across species, the same signature can exhibit different regulatory behaviours depending on the abundance and strength of each underlying mark.

In *U. pinna* females, genes assigned to signatures S12 and S14 indeed show no detectable expression despite being nominally "active". However, on closer inspection during our revisions, we realised that these signatures are extremely rare, particularly in *U. pinna*. These signatures are thus likely to represent

rare or unstable mark combinations rather than biologically meaningful categories. Such low-abundance states are a common outcome of chromatin-state modelling when uncommon combinations of histone marks are partitioned into separate states.. We now clarify these points more clearly in the revised manuscript.

The reviewer also noted the discrepancy between *U. pinnatifida* males and females. As discussed in the manuscript, the male tissue contained a high abundance of sperm cells, which likely resulted in a mixed chromatin landscape characterised by an elevated proportion of the null signature S1. This cellular heterogeneity is the most likely explanation why expression values for S12 and S14 differ between the two panels, which again, are also extremely rare combinations in the *U. pinnatifida* male sample. We now state this more explicitly in the revised manuscript.

4. Line 283-296, this part of the result, i.e., the comparison of chromatin states between sexes, is detached from previous results. Because emergence of dioecy or turnover of sex chromosomes occurred after loss of DNA methylation or gain of H3K79me function. Given their recent origin, there are many species specific pattern of chromatin states between sexes. For example, some species involve changes only in female-biased genes, some involve both male- and female-biased genes. However these descriptive patterns seem to lack explanations or hypotheses to be accounted for. I suggest author could refer to the works in plants (Zemp et al. 2016 Nat. Plants) and also their previous work of comparing expression changes between co-sexuality vs. dioecious algae (Cossard et al. 2022 Nat. Ecol. Evol) to build up more links between the results and explanations related to phenotypes/ecological traits of different species.

Re: We thank the reviewer for this insightful suggestion. Our reasoning in analysing the links between chromatin processes and sex determination in these species was to provide a concrete biological example of how the derived chromatin toolkit of brown algae regulates development. We have now incorporated the reviewer's suggestion and revised the Discussion to better integrate our chromatin-based observations with existing work on sex-biased gene regulation and the evolution of sexual systems in other lineages. Specifically, we now draw explicit parallels to the suggested studies in *Silene latifolia* (Zemp et al., 2016) and to transcriptomic analyses of transitions between dioicy and co-sexuality in brown algae (Cossard et al., 2022). These additions clarify how the species-specific chromatin reconfigurations we report are consistent with broader evidence that regulatory evolution underlying the diversity and evolutionary transitions of sexual systems is often modular and lineage-specific.

*****END*****

Reviewer #1:

Remarks to the Author:

The authors have convincingly addressed my previous comments, and I believe that the revisions have strengthened the manuscript overall. The improved formatting of the figures and extended data figures is effective in maximising clarity, and I also appreciated the expanded discussion of U/V sex chromosomes in other systems. I agree with the authors that, in its current form, the association between hPTMs and TE classes is not especially informative. A comprehensive, species-specific annotation of TEs would be required to address this properly, which would clearly represent a substantial effort and is more appropriate for a potential future dedicated study. While my initial hope was that the curated Ectocarpus TE library might allow identification of sufficiently abundant or young TEs in the other species, it is clear that this group is highly diverse, having diverged approximately 450 million years ago. In this context, I agree with the authors that these preliminary analyses are better omitted from the main manuscript. The same reasoning applies to the viral insert analyses. I have only one minor remaining comment:

Line 619: While H3K36me2 has been shown to recruit DNMT3A in mammals (PMID: 31485078), this mark is predominantly intergenic rather than a gene-body modification. The authors may instead wish to refer to H3K36me3, which has been shown to correlate with gene-body 5mCG in both vertebrates and invertebrates.

We thank Dr de Mendoza for his support of our article and the critical reading of our initial manuscript. His suggestions and comments have helped to greatly improve the clarity of our study. We have now included these other references as suggested.

Alex de Mendoza

Reviewer #2:

Remarks to the Author:

The motivation to look into a role of H3K79me2 over gene bodies seems clearer to me now. To illustrate to what extent this mark contributes to gene repression across all genes in the genome, I think it is important that the authors mention in the text that they perform the analyses over a subset of genes (those that overlap H3K79me2 peaks as stated in the response letter). They should also add for each species the number of genes examined the same way as for *S. ischiensis* (753/21187; 3.5%). Along these lines and following up on their discussion on a potential contribution of this mark in repressing transposable elements, is it possible that genes are not specifically targeted by H3K79me2, but rather that the mark accumulates over genes due to their proximity to repressed transposable elements? If the authors agree with this hypothesis, a short discussion could be added to the manuscript.

Firstly, may we thank Reviewer 2 for their questions and comments, as these have helped to greatly improve the quality and clarity of our study.

In addition to our comment on line 157 stating how H3K79me2 marks “*approximately 30% - 50% of genes depending on the species*”, we have now added the precise number of genes overlapping an H3K79me2 peaks to Figure 2E for further

transparency. We have also added the following comments to the figure legends for further clarity:

Figure 2E legend: “*The metaplots are restricted to genes that overlapped with an H3K79me2 peak*”

Extended Data Figure 3 legend: “*Unlike in Figure 2E, which displays only genes overlapping an H3K79me2 peak, these metaplots include all genes across the genome of each species.*”

Our previous response to Reviewer 2 may give a more detailed answer in regard to the potential role of H3K79me2 in repressing transposable elements. Our work in *Ectocarpus* shows that H3K79me2 and sRNAs have a small but significant preference for intact TEs, although no strong enrichment is observed for most classifiable TE families (Dinatale et al., 2025, *Genome Biology*). We once again looked at intact young TEs across species (defined as long TEs with <20% divergence from their consensus sequence) and again found no strong preference of H3K79me2 for specific TE classes. We thus think that it is unlikely H3K79me2 is lured to genes by neighbouring TEs and as a such, we have not added these comments to the revised manuscript.

Other comments:

Line 194: I guess it should be “transcription start site” instead of “translational start site”

Unfortunately transcription start sites remain unannotated in these brown algal genomes, so we instead used translational start sites (ATG) as the anchoring point in our analysis. We have now added this point to the methods section for further clarity.

Reviewer #3:

Remarks to the Author:

The authors have addressed all my previous questions and comments with satisfaction. I don't have further questions

We would like to thank Reviewer 3 for their comments, which helped to greatly improve the quality of our study.